# Observations of turbulent mixing in the Dotson Ice Shelf cavity

Maren Elisabeth Richter[1], Karen J. Heywood[1], Rob A. Hall[1], and Peter E.D. Davis[2]

[1]Centre for Ocean and Atmospheric Sciences, School of Environmental Sciences, University of East Anglia, Norwich, NR4 7TJ, United Kingdom
[2]u i

**Correspondence:** Maren Elisabeth Richter (m.richter@uea.ac.uk)

**Abstract.** Dotson Ice Shelf (DIS) is located in the Amundsen Sea sector of Antarctica, an area of rapid glacial mass loss due to ocean-driven basal melting. Here, warm Circumpolar Deep Water is transported onto the continental shelf and can access ice shelf cavities and deep grounding lines, causing melting, glacial retreat and thus sea level rise. The circulation of this warm water, and the heat transport within ice shelf cavities, remains mostly unknown. We collected data from over 100 km of dive tracks along the seabed under DIS using an autonomous vehicle, AutoSub Long Range. This study presents observations of ocean velocity, turbulent kinetic energy dissipation rate ($\varepsilon$) from microstructure measurements, and heat flux calculations. Rates of background mixing are $\varepsilon \approx 10^{-10}$ W kg$^{-1}$ with patches of higher mixing of $\varepsilon \approx 10^{-8}$ W kg$^{-1}$. Higher turbulent kinetic energy dissipation rate is associated with stronger along-slope currents, high vertical current shear, steeper bathymetry, and positive temperature anomalies. Average vertical heat fluxes are on the order of $0.1$ W m$^{-2}$ and maximum heat fluxes reach $52$ W m$^{-2}$. This is compared to the $59$ W m$^{-2}$ to $176$ W m$^{-2}$ needed to maintain observed average basal melt rates at DIS. Turbulent mixing is higher in the fast-flowing inflow region and over rough topography. We show a highly complex spatial pattern of turbulent mixing and of bottom topography. The bottom topography is currently not resolved in bathymetry products and both the topography and turbulent mixing are currently not resolved in models of ice-shelf–ocean interactions. The levels of turbulent mixing experienced by the warm mCDW inflow to the DIS will lead to negligible loss of heat during its path to the grounding line, leaving plenty of heat available to melt the ice shelf base there. Higher average vertical heat fluxes than observed here must occur in areas of the cavity not accessed in this study.

## 1 Introduction

Dotson Ice Shelf (DIS) is located on the southern boundary of the Amundsen Sea, an area where the West Antarctic Ice Sheet is losing mass, largely driven by increasing ocean heat flux toward and beneath ice shelves. DIS contributes disproportionately to the total Amundsen Sea ice mass loss (Rignot et al., 2019). Between 1979 and 2017 DIS contributed $0.6$ mm to global eustatic sea level rise (Rignot et al., 2019). The rate of discharge across its grounding line has increased throughout the satellite record (Rignot et al., 2019; Mouginot et al., 2014) and the grounding line has retreated (Rignot et al., 2014; Scheuchl et al., 2016; Milillo et al., 2022). The increased ice flux across the Dotson grounding line, coupled with the stable ice flux across the calving front (Rignot et al., 2013; Mouginot et al., 2014) and the increased thinning of the ice shelf (Rignot et al., 2013; Mouginot et al., 2014; Gourmelen et al., 2017; Greene et al., 2022) leads to the conclusion that ocean thermal forcing has increased basal melt

of the ice shelf (e.g. Mouginot et al., 2014). Dotson has thinned at a 37% higher rate than the average rate of thinning in the Amundsen Sea (Paolo et al., 2015). Basal melt in the Amundsen Sea is driven by the intrusion of warm modified Circumpolar Deep Water (mCDW) onto the continental shelf where it can flow into ice shelf cavities. The mCDW can cause melting at the grounding line, leading to basal mass loss and grounding line retreat. It has been suggested that there is a strong seasonality in the velocity and heat content of the inflow to the DIS cavity and in the velocity and meltwater content of the outflow, with maximum inflows in summer and maximum outflows in autumn (Yang et al., 2022). This seasonality, as well as interannual variability at DIS, are hypothesised to be driven by local winds and sea-ice conditions (Yang et al., 2022; Kim et al., 2021).

Basal melt under Dotson is highest close to the grounding line of the Kohler East (often referred to as Smith West) and Kohler West glaciers (Khazendar et al., 2016; Gourmelen et al., 2017). The Kohler West grounding line lies at the southern end of the dashed path shown in Figure 1a. A cross-section of the cavity along the path (Figure 1b) shows an idealized view of the cavity circulation under the Dotson Ice Shelf. Warm water entering the cavity in the east, and traveling along a path shallower than the 830 m deep sill (Jordan et al., 2020), can reach the grounding line. Warm water that reaches the grounding line causes high basal melt and grounding line retreat (Khazendar et al., 2016; Gourmelen et al., 2017). The sill may limit direct access of the deepest and warmest mCDW to the grounding line (Jordan et al., 2020; Khazendar et al., 2016). The addition of meltwater to the warm, salty mCDW forms a buoyant current which travels along the underside of the ice before exiting the cavity in the west. Along its path, the water experiences turbulent mixing which can transport heat and salt upward, modifying the properties of both the inflowing water, which ultimately interacts with ice near the grounding line, and water carried by the buoyant current out of the cavity.

Input of meltwater from ice shelves influences the local and global ocean circulation and climate (Hellmer, 2004; Silvano et al., 2018; Bronselaer et al., 2018), as well as sea-ice formation and persistence (Hellmer, 2004; Richardson et al., 2005; Bintanja et al., 2013; Bronselaer et al., 2018). Thus, it has important effects on ocean heat and carbon storage and transport (Silvano et al., 2018). Increased meltwater input leads to decreased winter mixing and thus to easier access of warm deep water to the Amundsen Sea ice shelves, causing further melting (Silvano et al., 2018). All these effects are modulated by the depth at which meltwater is injected into the ocean. The depth at which meltwater enters the ocean is influenced by where melt predominantly occurs. For DIS, melt occurs at the grounding line (Khazendar et al., 2016), in outflow channels along the underside of the ice shelf (Gourmelen et al., 2017), and in locations spread inhomogeneously over the entire ice shelf base in highly complex patterns (Wåhlin et al., 2024a). Flow in the western DIS is intensified at the ice base with observed melt rates consistent with shear driven turbulence and heat transport. The central and eastern areas of DIS show low flow speeds close to the ice base and low meltwater concentrations (Wåhlin et al., 2024a). The spatial distribution of ice shelf melt has an important effect on ice shelf stability: melting concentrated at the grounding line leads to stronger grounding line retreat (Walker et al., 2008); and melting concentrated in channels may lead to weakening and break up of the floating ice shelf (Gourmelen et al., 2017).

The input of meltwater to the Amundsen Sea is also important for biological activity in the region. The flow of mCDW along the seafloor on its way into the DIS cavity enriches the mCDW in dissolved iron and manganese while the meltwater from the ice shelf itself is a source of particulate iron and manganese (van Manen et al., 2022). The addition of glacial meltwater makes

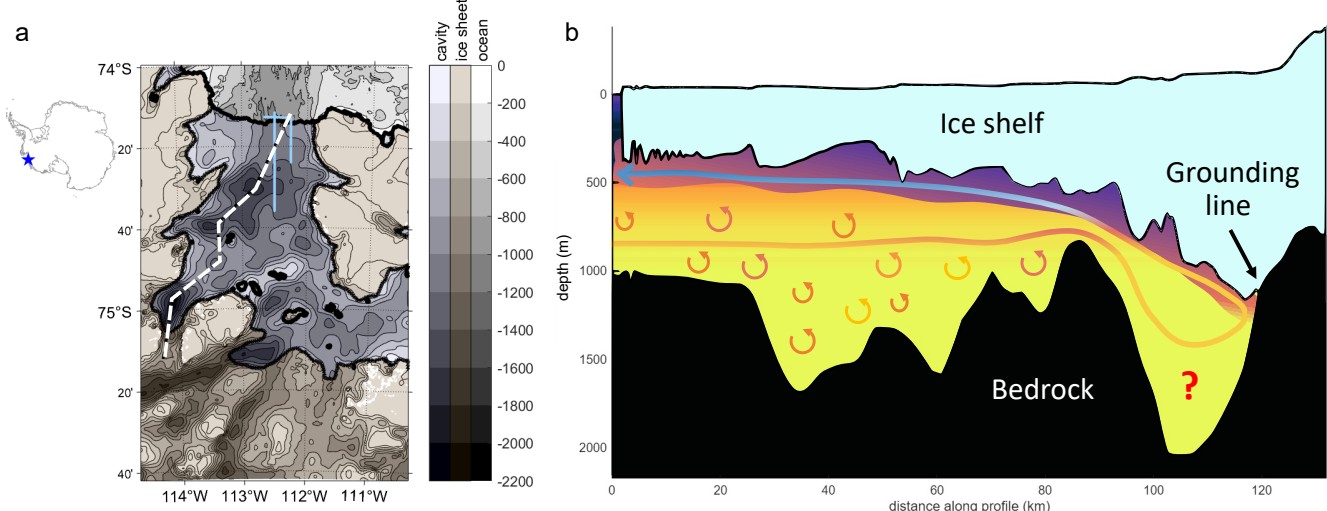

**Figure 1.** Schematic of the Dotson Ice shelf cavity and the circulation within. a: map of the bedrock depth under and around the Dotson Ice Shelf. The dashed white line runs along the deepest route to the Kohler west grounding line (Jordan et al., 2020). The blue lines show the dive tracks of the ALR. The inset to the left shows the location of Dotson Ice Shelf in Antarctica. b: cross-section of the Dotson Ice Shelf cavity along the dashed line in a. The cross-section shows a cartoon of the possible pathway for warm mCDW to the grounding line and the buoyant meltwater current exiting the cavity. Turbulent mixing can alter water properties along the water path. The bathymetry and ice shelf geometry in a and b is from BedMachine v3 (Morlighem, 2022).

the outflowing mCDW more buoyant than the dense mCDW inflow, transporting iron and manganese to the surface ocean (van Manen et al., 2022) where they are important micronutrients for primary producers (Twining and Baines, 2013).

Melt rates can be highly variable in time and space, even under a single ice shelf (e.g. Davis et al., 2018; Gourmelen et al., 2017). To study where heat from mCDW interacts with the ice shelf and thus releases meltwater, we need to understand the transport and mixing processes between mCDW and overlying colder and fresher water masses within the cavity. The turbulent kinetic energy dissipation rate, $\varepsilon$, is the rate at which molecular viscosity dampens isotropic turbulence generated at large scales by e.g. vertical or lateral shear, and is used to quantify turbulent mixing. Due to the remote location and difficult access, measuring turbulent kinetic energy dissipation rate in ice shelf cavities is only now starting to become feasible. To our knowledge, there exist two published studies of mixing in an ice-shelf cavity measured by an underwater vehicle, one under Pine Island Glacier (Kimura et al., 2016), and one under the Filchner Ronne Ice Shelf (Davis et al., 2022). We present a third such study, targeting DIS. DIS and Pine Island Ice Shelf experience low tidal flows, whereas Filchner Ronne Ice Shelf experiences strong tidal flows. Unlike Davis et al. (2022) and Kimura et al. (2016), our study targets the current of warm mCDW flowing into the ice shelf cavity and maintains a dive track close to the seabed. We investigate the circulation and mixing in the mCDW inflow close to the bed of the cavity to understand the effect of bathymetry on mixing and circulation. We quantify the upward heat transport that cools the mCDW in the deepest part of the cavity whilst warming the overlying mCDW (which can access the grounding line and the ice shelf base; Figure 1), and investigate drivers for the observed mixing.

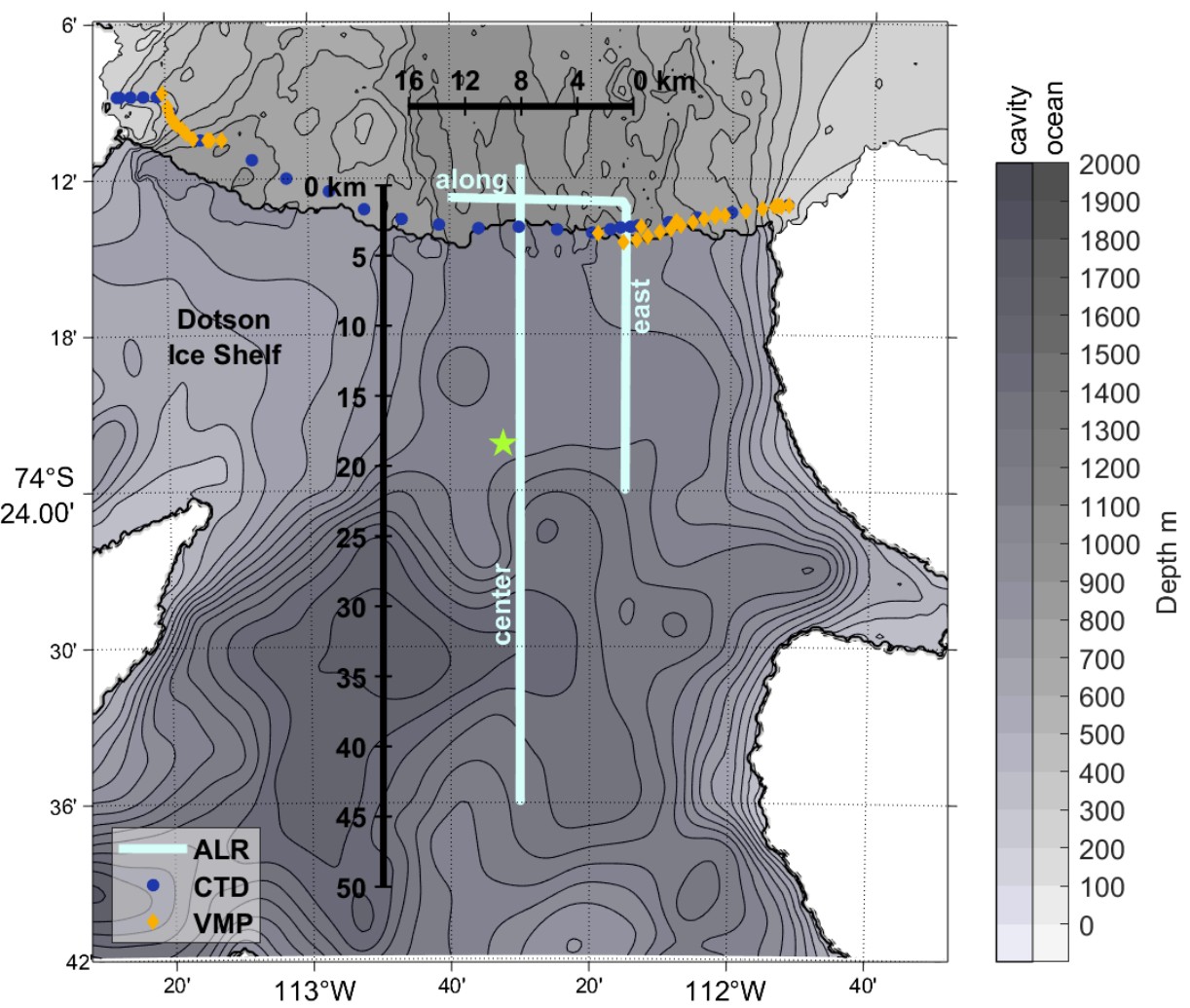

**Figure 2.** Map of Dotson Ice Shelf cavity and surroundings, with locations of data sets. AutoSub Long Range (ALR) dive tracks (light blue), Conductivity-Temperature-Depth (CTD) casts (dark blue circles), Vertical Microstructure Profiler (VMP) casts (orange diamonds), and the location of a CTD cast through the DIS into the cavity below (green star). Bathymetry is from Bedmachine V3 (Morlighem, 2022), with different grey scales for depths within and outside the cavity. The black line denotes the ice front.

| dive | start time | end time | duration (h) | distance (km) |
|------|-----------|----------|--------------|---------------|
| centre_short | 21.01.2022 06:50 | 22.01.2022 00:27 | 17.6 | 41 |
| centre_long | 02.02.2022 05:46 | 04.02.2022 01:06 | 43.3 | 91 |
| east | 05.02.2022 07:17 | 06.02.2022 02:40 | 19.4 | 41 |
| along | 06.02.2022 07:55 | 06.02.2022 14:05 | 6.2 | 13 |

**Table 1.** Metadata for the four successful dives performed by the ALR. Distance is the total length of the track summing both into and out of the cavity (or the length of the dive track along the ice front for the "along" mission).

## 2 Data and methods

In 2022 DIS was surveyed with Conductivity, Temperature, Depth profilers (CTD), Lowered Acoustic Doppler Current Profilers (LADCP) and Vertical Microstructure Profilers (VMP) at the ice front and several dive tracks into the ice shelf cavity using an AutoSub Long Range (ALR) underwater vehicle (Figure 2). The ALR travelled along the bottom of the cavity, keeping a distance of $100\,\mathrm{m}$ from the seabed. It was equipped with a Rockland Scientific International (RSI) MicroRider carrying two microstructure shear probes and two fast-response thermistors. It also carried an upward and a downward looking ADCP and Seabird Electronics temperature, conductivity, pressure, and oxygen sensors. The ALR performed four successful missions (Table 1). Of these, all except the centre_long mission had successful microstructure measurements. Centre_long does however provide valuable information on currents and hydrography in the central cavity. A clock offset of approximately 2 minutes between the ALR CTD and the MicroRider was resolved by calculating lagged correlations between the MicroRider pressure sensor and the CTD pressure sensor to find the offset, then correcting for the identified clock offset and drift.

The ALR ADCP data were collected at $1\,\mathrm{s}^{-1}$ frequency in twelve vertical $8\,\mathrm{m}$ bins and were processed with modified code from Eleanor Frajka-Williams' GitHub page (https://github.com/eleanorfrajka/alr_processing_dynopo), which builds on Rick Pawlowicz's RDADCP functions (https://www-old.eoas.ubc.ca/~rich/#RDADCP). The ADCP delivered good quality measurements up to $50\,\mathrm{m}$ from the ALR. To decrease noise in the ADCP data, and to make displaying the data easier, we show 2-minute and 10-minute temporal medians of the ADCP velocities. In cases where information on the vertical shear is not required, we additionally took a vertical median of the ADCP bins. A spectral analysis of the ADCP current velocities showed no signal at tidal frequencies, thus, we chose not to detide them. Further, the Bedmachine V3 (Morlighem, 2022) bathymetry product used to generate the CATS2008 v2023 tide model (Erofeeva et al., 2024) show water depths several hundred metres different to those measured by the ALR. We calculated the directional gradient of the Bedmachine V3 bathymetry to derive across and along isobath velocity components, and provide a measure of the steepness of the topography. For this, we upsampled the Bedmachine bathymetry from its native $500\,\mathrm{m}$ resolution (Morlighem, 2022) to a denser $10\,\mathrm{m}$ resolution by cubic interpolation. This resulted in a smooth bathymetry gradient which was then linearly interpolated onto the ALR dive track.

The microstructure data from the ALR were processed to derive turbulent kinetic energy dissipation rate ($\varepsilon$) with the ODAS v4.5.1 toolbox provided by RSI, following the best practices published by Lueck et al. (2024) and used by Davis et al. (2022)

for their ALR mission. Turbulent kinetic energy dissipation rate is derived from shear variance using Equation 1

$$\varepsilon = \frac{15}{2}\nu\overline{\left(\frac{\partial v}{\partial x}\right)^2}. \tag{1}$$

$\nu$ is the temperature-dependent molecular kinematic viscosity of water, $\overline{\left(\frac{\partial v}{\partial x}\right)^2}$ is the variance of the velocity shear fluctuations along the path of the ALR (Osborn, 1974; Oakey, 1982). We converted raw velocity shear into physical units using the ALR speed through water ($\sim 0.6\,\mathrm{m\,s^{-1}}$), derived from the ALR ADCP water track and bottom track velocities. The shear data and the accelerometer signal recording the vehicle vibrations were high-pass filtered with a Butterworth filter, forward and backward, with a cut-off frequency of $0.25\,\mathrm{Hz}$. We calculated spectra of velocity shear over $4\,\mathrm{s}$ segments of data with $2\,\mathrm{s}$ overlap between segments. To increase signal-to-noise-ratio we averaged shear spectra over half-overlapping $32\,\mathrm{s}$ windows. This results in a turbulent kinetic energy dissipation rate every $32\,\mathrm{s}$ seconds along the ALR track equivalent to every $19.2\,\mathrm{m}$. The effect of vehicle vibrations was removed by applying the Goodman method (Goodman et al., 2006) to the shear and accelerometer data. This removes the signal in the shear spectrum that can be related to the accelerometer signal. Unlike microstructure measurements performed with a small, light-weight AUV (e.g. Kolås et al., 2022), the shear microstructure recorded on AutoSub Long Range was not critically impacted by vehicle vibrations, possibly due to its greater mass. The shear power spectra from a MicroRider mounted on an ALR have been described in detail in Davis et al. (2022). Broad peaks in the power spectrum of the accelerometer signal caused by vehicle motion and the AUV propeller occur at frequencies above $10\,\mathrm{Hz}$, frequencies higher than the frequencies at which the Nasmyth spectra (Nasmyth, 1970), fitted to the power spectra of the shear, roll off. Smaller, narrower peaks at frequencies below $10\mathrm{Hz}$ in the accelerometer spectra are successfully removed by the Goodman method for dissipation rates above $1 \times 10^{-8}\,\mathrm{W\,kg^{-1}}$. Deviations from the fitted Nasmyth spectra remain for dissipation rates below $1 \times 10^{-9}$, arguing that quantitative estimates of dissipation rate in very quiescent regimes are not as reliable as estimates of high dissipation rates. Individual dive tracks show good agreement between shear spectra and Nasmyth spectra for dissipation rates lower than $1 \times 10^{-10}\,\mathrm{W\,kg^{-1}}$. Where dissipation rates calculated from two orthogonal shear probes show good agreement, we are confident in reporting dissipation rates down to $1 \times 10^{-11}\,\mathrm{W\,kg^{-1}}$. Additionally, any signal in the shear spectra caused by the AUV motion, and not removed by the Goodman filter, will have minimal effects on the spatio-temporal pattern of high and low $\varepsilon$ observed by the ALR or the qualitative assessment of these patterns, on which this study focuses.

Turbulent diapycnal diffusivity $\kappa$ is a measure of the vertical mixing of heat and mass and was estimated following Osborn (1980) as

$$\kappa = \Gamma\frac{\varepsilon}{N^2} \tag{2}$$

where $\Gamma = 0.2$ is the mixing efficiency, a measure of the amount of available turbulent kinetic energy that is permanently converted to potential energy by turbulent mixing, which is generally set to $0.2$ (Osborn, 1980). The Brunt-Väisälä frequency $N^2$ was calculated from the vertical density gradient below $900\,\mathrm{m}$ (the depth range occupied by the ALR) recorded in a CTD cast through a drill hole through DIS (Wåhlin et al. (2024a, b), about $15\,\mathrm{km}$ from the ice front; see green star in Figure 2 for location). The CTD cast was recorded on 7 February 2022, within 4 and 17 days and about $1\,\mathrm{km}$ to the west of the two

central ALR dive tracks. A constant $N^2$ was assumed because the density gradient is approximately linear at that depth, and also because vertical density profiles in the cavity and at the ice front are similar (Figure 4).

Vertical heat $Q_T$ and salt $Q_S$ fluxes were calculated using Equation 3:

$$Q_T = -\rho_0 C_p \kappa \frac{\partial T}{\partial z}; \quad Q_S = -\rho_0 \kappa \frac{\partial S_A}{\partial z}; \tag{3}$$

where $\rho_0$ is the potential density, $C_p$ is the specific heat capacity of seawater ($3992\,\mathrm{Jkg^{-1}K^{-1}}$), $T$ and $S$ are the Conservative Temperature and Absolute Salinity, respectively, and $\frac{\partial}{\partial z}$ denotes the vertical gradient.

Temperature changes $\Delta T$ of a seawater layer of thickness h over time t were calculated using Equation 4:

$$\Delta T = \left( \frac{Q_T}{\bar{\rho} C_P h} \right) \Delta t, \tag{4}$$

with $\bar{\rho} = 1028\,\mathrm{kg\,m^{-3}}$ a representative density for deep water in the DIS cavity.

Following Dotto et al. (2025) we assessed different turbulent mixing metrics in addition to $\varepsilon$ and $\kappa$. We calculated Ertel's potential vorticity q using the approximation:

$$q \approx \left( f + \frac{\partial v}{\partial x} \right) N^2 - \left( \frac{\partial v}{\partial z} \right) \left( \frac{\partial b}{\partial x} \right), \tag{5}$$

where $f$ is the Coriolis parameter, $v$ the current velocity,

$$b = -g \frac{\bar{\rho} - \rho}{\rho}, \tag{6}$$

is the buoyancy, $g$ is the gravitational acceleration, $f$ is the planetary vorticity, $\rho$ is the in situ density, and $\bar{\rho} = 1028\,\mathrm{kg\,m^{-3}}$ is the reference density. When calculating $q$ for the ALR data, we used the vertical distance between the good-quality bin closest to the ALR in the upward and downward looking ADCP data as $\Delta z$ (approximately $38\,\mathrm{m}$), and the horizontal distance between successive two-minute medians of each bin (approximately $72\,\mathrm{m}$) as $\Delta x$. The along-slope velocity component from these bins is $v$. For the section at the DIS front, we used the horizontal distance between neighbouring CTD casts as $\Delta x$, the

vertical resolution of the LADCP ($8\,\mathrm{m}$) as $\Delta z$, and the meridional component of the current velocity as $v$. We then calculated the Rossby number, Ro

$$\mathrm{Ro} \approx \frac{1}{f} \left( \frac{\partial v_{as}}{\partial x} \right) \tag{7}$$

with the choice of $v_{as}$ as the along-slope component of the velocity (for the currents measured by ALR) or as the northward velocity (for the ice front transect), as detailed above. Ro quantifies the role of Earth's rotation relative to the vertical component

of the relative vorticity.

We used $N^2$ and the shear measured by the ALR to calculate the dimensionless gradient Richardson number (Ri):

$$\mathrm{Ri} = \frac{N^2}{\left( \frac{\partial u}{\partial z} \right)^2 + \left( \frac{\partial v}{\partial z} \right)^2}, \tag{8}$$

where $u$ is the zonal velocity component. Thus, Ri is calculated from a constant value of $N^2$, based on a single profile in the cavity, and shear is a function of space and time along the track of the ALR. Variations of Ri due to variations in $N^2$ are not

captured. For constant $N^2$, Ri is low in areas of high shear. $\text{Ri} < 1/4$ is a necessary condition for turbulence generated by velocity shear (Hazel, 1972; Miles, 1961; Howard, 1961). Our values of Ri are biased high because the ADCP underestimates vertical shear (Polzin et al., 2002), thus we will confine our discussion of Ri to relative values.

    Instabilities can be categorised as gravitational, symmetric or centrifugal (Thomas et al., 2013). These instabilities occur when $q$ has the opposite sign to $f$. Gravitational, symmetric and centrifugal instabilities convert convective available potential

energy, vertical and lateral shear, respectively, into kinetic energy (Haine and Marshall, 1998). Following Thomas et al. (2013) we calculated $\phi_{Ri}$ and $\phi_c$ using Equation 8:

$$\phi_{Ri} = \arctan\left(-Ri^{-1}\right) \quad \text{and} \quad \phi_c = \arctan\left(-1 - \frac{1}{f}\left(\frac{\partial v_{as}}{\partial x}\right)\right). \tag{9}$$

    Since the water column shows no density inversions in our CTD section at the DIS front and the CTD cast through the DIS, we do not observe gravitational instability. The critera for symmetric instability are

$\phi_{Ri} < \phi_c \quad \text{and} \quad -90° < \phi_{Ri} < -45° \quad \text{and} \quad N^2 > 0 \quad \text{and} \quad \dfrac{\partial v}{\partial x}\dfrac{1}{f} < 0,$

or

$-90° < \phi_{Ri} < \phi_C \quad \text{and} \quad N^2 > 0 \quad \text{and} \quad \dfrac{\partial v_{as}}{\partial x}\dfrac{1}{f} > 0.$

    The criteria for centrifugal instability are

$-45° < \phi_{Ri} < \phi_C \quad \text{and} \quad N^2 > 0 \quad \text{and} \quad \dfrac{\partial v_{as}}{\partial x}\dfrac{1}{f} < 0.$

To support the measurements by the ALR, we additionally analysed microstructure profiles along the ice front measured with a RSI Vertical Microstructure Profiler (VMP). The VMP was deployed from a ship moving at $\sim 0.5\,\mathrm{m\,s^{-1}}$, with the VMP continuously profiling between the sea surface and $\sim 50\,\mathrm{m}$ above the seabed. The shear microstructure from the VMP was processed following Naveira Garabato et al. (2017). For details of the processing see Dotto et al. (2025). A ship-based hydrographic survey along the ice front provided temperature and salinity measured with a shipboard Seabird Scientific

Conductivity-Temperature-Depth (CTD) instrument and current velocity measured by a lowered Acoustic Doppler Current Profiler (LADCP). The CTD measurements were post-cruise calibrated and binned in 2-m vertical medians. Conservative Temperature and Absolute Salinity were calculated using the TEOS-10 toolbox (McDougall and Barker, 2011). Upward-looking and downward-looking LADCP measurements were processed with the LDEO_IX toolbox, incorporating information from the vessel-mounted ADCP, CTD, GPS and bottom track from the LADCP (Thurnherr, 2021). The processed data were

averaged into 8-m vertical bins and detided using an updated version of the CATS2008 Antarctic tide model (Padman et al., 2002; Erofeeva et al., 2024). Modelled tidal current components are on the order of $1\,\mathrm{cm\,s^{-1}}$ at the ice front and the tide model agrees well with tides extracted from the shipboard ADCP data (Dotto et al., 2025). Conversely, the ALR ADCP data are not detided due to the ill-constrained bathymetry under DIS, the absence of a detectable tidal signal in a spectral analysis of the ALR ADCP currents in the cavity, and the risk of degrading the ADCP data quality with an ill-fitting tidal model. Thus, we

use the best data processing available to us, both inside and outside of the cavity.

We compare our values of $\varepsilon$ to the literature by plotting kernel density distributions. We set the bandwidth of the kernel to $10^{0.2}\,\mathrm{W\,kg^{-1}}$ and the upper and lower cutoff values to the maximum/minimum of the distribution plus/minus $< 2 \times 10^{-10}\,\mathrm{W\,kg^{-1}}$.

## 3 Results and Discussion

### 3.1 The Dotson embayment and ice front

During our observations, the DIS front is characterized by relatively warm and fresh surface waters in the upper $100\,\mathrm{m}$, a layer of colder Winter Water (WW) at $100\,\mathrm{m}$–$400\,\mathrm{m}$, and warmer, saltier mCDW below that (Figure 3a). Below the WW temperature minimum, the water column is salt-stratified with fresher colder water overlying warm salty water. The temperature and salinity at the ice front are within the historic range of watermass distributions and properties at DIS (Kim et al., 2021). The ice shelf draft is approximately $300\,\mathrm{m}$ at the front, meaning that typically only mCDW can enter the cavity. A bottom intensified
southward current flows into the cavity in the east, between the $400\,\mathrm{m}$ and $900\,\mathrm{m}$ isobaths, and a shallower, bottom intensified northward current flows out of the cavity in the west (Figure 3c). The outflow region is not the focus of this paper and is discussed in detail by Dotto et al. (2025). The ALR measurements just skim the western edge of the inflow. Below $500\,\mathrm{m}$ depth, turbulent kinetic energy dissipation is elevated in the inflow (compared with other areas below $500\,\mathrm{m}$ along the ice front). Turbulent kinetic energy dissipation is $\approx 10^{-8}\,\mathrm{W\,kg^{-1}}$ in the inflow over an area approximately $7\,\mathrm{km}$ wide and $200\,\mathrm{m}$
high (Figure 3d; turbulent kinetic energy dissipation rate is elevated between 38 km and 45 km of the ice front and $\sim 200\,\mathrm{m}$ above the seabed). The centre of the ice front did not have VMP measurements, however, the ALR and the VMP measurements farthest from the lateral walls suggest that turbulence is low here. In the area where we have both VMP and ALR measurements of turbulent kinetic energy dissipation rate, both instruments record values of the same order of magnitude. This means that the ALR measurements under DIS can be interpreted with confidence as comparable in quality to the VMP measurements.

We investigate the stability of the flow at the ice front (Figure 3). Instabilities may develop when potential vorticity and $f$ have opposite signs, as $f$ is negative in the southern hemisphere, potential vorticity $> 0$ indicates conditions favourable to instability. At the DIS front, potential vorticity $q$ is almost uniformly negative, with very low values above and below the Winter Water at $200\,\mathrm{m}$ and $400\,\mathrm{m}$ depth (Figure 3e). This is mainly driven by the strong stratification above and below the WW layer (Figure 3i). The inflow and outflow regions show high absolute values of Ro (Figure 3g) indicating high lateral shear. Measures
of vertical and horizontal current shear are generally lower in the inflow than in the outflow (Ro and shear$^2$ in Figure 3g+j). Following Thomas et al. (2013), we classify instabilities as symmetric (in the inflow region) and as symmetric and centrifugal (in the outflow region) (Figure 3f). Because $N^2$ is positive along the entire ice front transect (Figure 3i), we do not observe gravitational instability. Symmetric instability is driven by high vertical current shear (Figure 3j). The region of high turbulent kinetic energy dissipation rate $\varepsilon$ in the inflow (Figure 3d) coincides with instances of low Ri captured at $40\,\mathrm{km}$ (Figure 3h).
Turbulent kinetic energy dissipation rate is larger than $10^{-8}$ here, one to two orders of magnitude higher than the background value (Figure 3d). Dotto et al. (2025) found similar results for the outflow of DIS. Although areas of high $\varepsilon$ extend beyond areas of low Ri, $\varepsilon$ is higher and Ri is lower in the upper watercolumn and close to the seabed. We observe areas of low Ri that are not associated with high values of $\varepsilon$, e.g. at $25\,\mathrm{km}$ along the transect. Here, in the centre of the DIS front we observe a

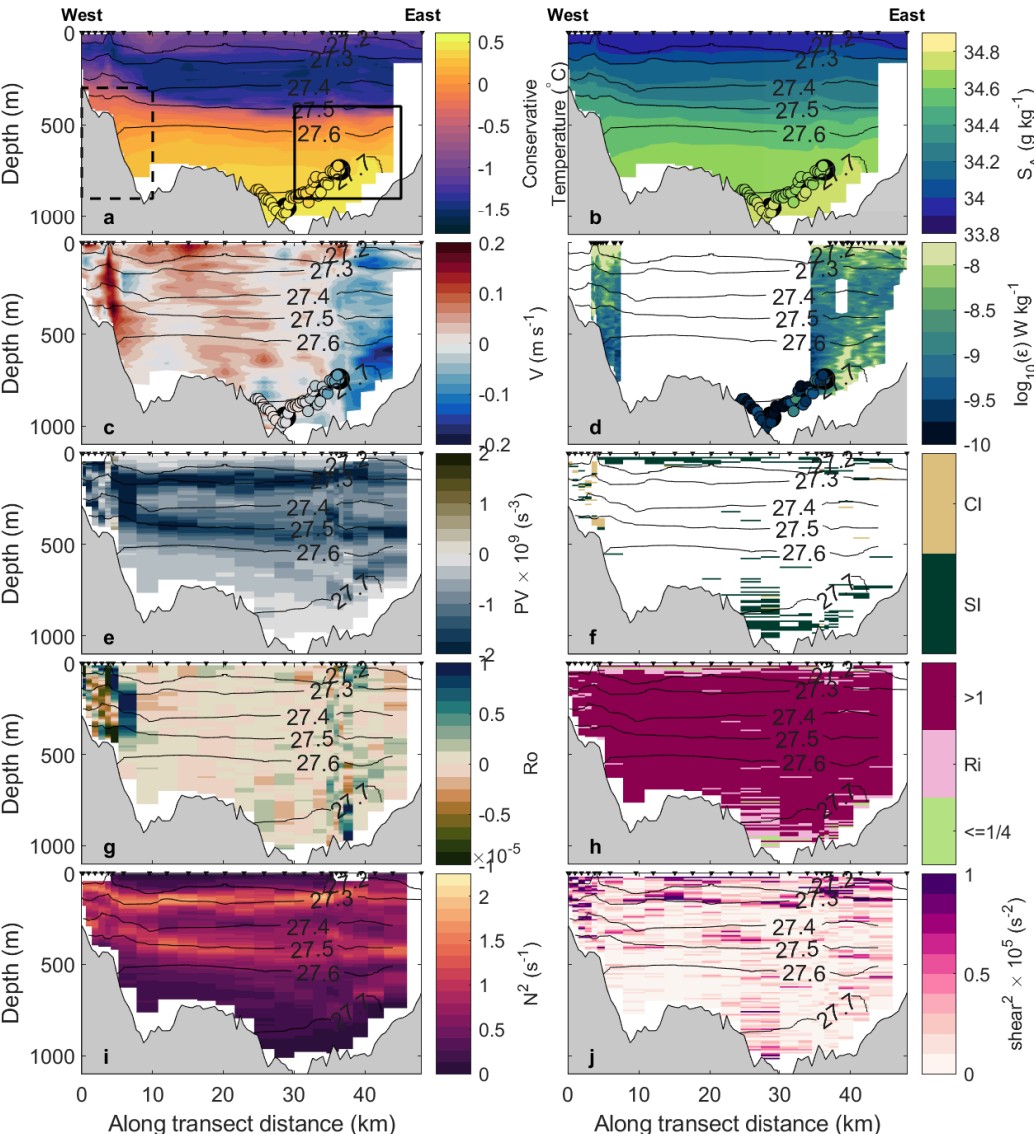

**Figure 3.** Sections along the DIS front of Conservative Temperature (a); Absolute Salinity (b); meridional velocity (c); turbulent kinetic energy dissipation rate ($\varepsilon$ (W kg$^{-1}$) (d); Ertel's potential vorticity $q$ (e); classification of instability following Thomas et al. (2013), SI = symmetric instability, CI = centrifugal instability (f); Rossby number (g); gradient Richardson number (h), values $< 1/4$ indicate conditions favourable for mixing driven by vertical shear, values $> 1$ indicate conditions not favourable for shear driven mixing; Brunt-Väisälä frequency (i); vertical shear squared (j). The view is out of the cavity, distance is from the western edge of the transect. Black contours show potential density. The small triangles at the top of the panels show the location of the measurements. 10-minute medians of the values measured by the ALR are shown as coloured dots in panels a-d. The two dots with bold outlines show the starting locations of the ALR east and centre short dive tracks into the cavity. The bathymetry is from Bedmachine V3 (Morlighem, 2022). The solid and dashed outline in (a) show the areas over which the inflow and outflow temperatures were averaged for Equation 11.

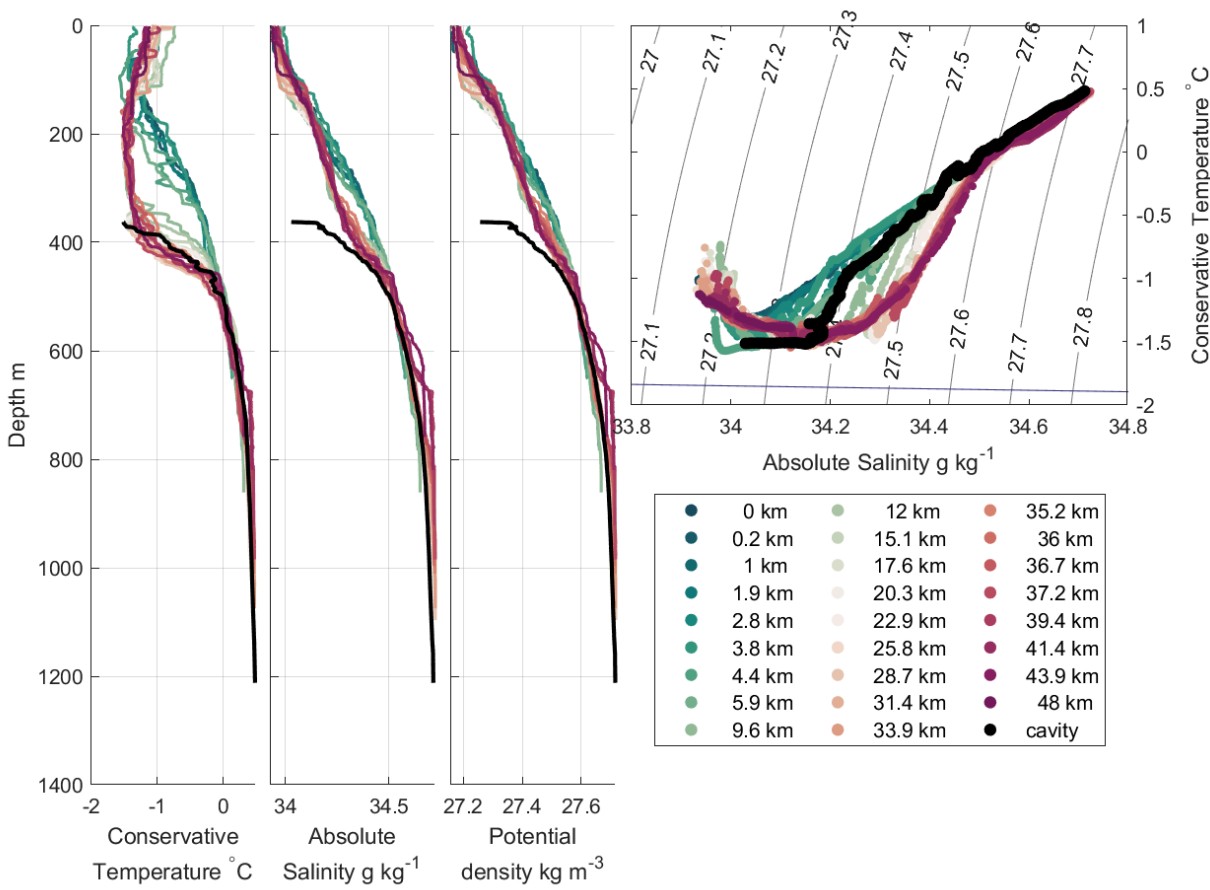

**Figure 4.** CTD profiles from the section along the ice front as coloured lines and from the mooring location in black (see Figure 2 for locations). Profiles in the inflow region (farther east) are in reds, profiles from the outflow region (farther west) are in greens. (a) Conservative temperature, (b) Absolute salinity, (c) potential density and (d) Conservative Temperature - Absolute Salinity diagram with isopycnals in grey and freezing line in blue.

small region of southward current and elevated vertical and lateral current shear. We do not have VMP measurements in the centre of the DIS front and thus can not say with certainty what the turbulence response is. The ALR did not observe a strong southward flow or high turbulent kinetic energy dissipation rate close to the seabed here. Because the ALR measurements were not coincident in time with the LADCP section, the ALR may have failed to capture transient patches of high turbulent kinetic energy dissipation rate present in the LADCP section. Nonetheless, low values of the Richardson number and the conditions conductive to symmetric instability suggest that there may be areas of high turbulent kinetic energy dissipation outside of the main inflow and outflow branches. These might have noticeable local effects (e.g. stirring up sediment) but are unlikely to influence the major circulation within the cavity. Overall, our observations show turbulent mixing to be patchy, bottom intensified and to coincide with high velocities (Figure 3c + d).

At the nearby Pine Island Ice Shelf (PIIS) Naveira Garabato et al. (2017) conducted ADCP and VMP transects along the calving front. Naveira Garabato et al. (2017) do not detect a fast, narrow, turbulent inflow current, unlike what we observed at DIS (Figure 3). High rates of turbulent kinetic energy dissipation below the WW were mostly confined to the PIIS outflow. The PIIS is connected to another ice shelf cavity to the north and may receive some of its inflow from under this neighbouring ice shelf, which may decrease the inflow across the PIIS front and possibly the turbulent mixing there. Additionally, the ice shelf draft of the PIIS is deeper ($\approx 400\,\mathrm{m}$) than the DIS ($\approx 350\,\mathrm{m}$). The ice shelf draft induces an abrupt change in water column thickness, blocking flow along isolines of water column thickness, and thus limits barotropic inflow to the cavity (Wåhlin et al., 2020), thus decreasing inflow current velocities and possibly turbulent mixing.

## 3.2 ALR and CTD observations in the ice-shelf cavity

### 3.2.1 Hydrography in the cavity

The CTD profile through the DIS at the mooring location (Figure 4, black line) displays an approximately $100\,\mathrm{m}$ thick layer with reduced salinity and density close to the ice shelf base. Below $600\,\mathrm{m}$, the water column properties inside and out of the cavity are very similar (Figure 4). The temperature profile taken through the DIS closely matches the temperature profiles in the inflow region at the ice front. We estimate $N^2$ below a depth of $900\,\mathrm{m}$ to be $6 \times 10^{-7}\mathrm{s}^{-1}$. This is about three orders of magnitude lower than typical open ocean values for the southern ocean (King et al., 2012), indicating weakly stable stratification in the cavity. Our ALR measurements capture the western flank of the inflow into the DIS cavity, as well as some of the outflow (Figure 3). In the cavity, the ALR detected currents that flow predominantly southeastward with low vertical shear in the east dive track, and a more mixed pattern in the two centre dive tracks (Figures 5 and 6). Current speeds in the cavity mostly ranged between $0.03\,\mathrm{m\,s}^{-1}$ and $0.04\,\mathrm{m\,s}^{-1}$, with maximum current speeds up to $0.11\,\mathrm{m\,s}^{-1}$ (Table 2). Current directions show evidence of bathymetric steering (Figure 5). This is particularly evident in the east dive track and around $25\,\mathrm{km}$ along the centre dive tracks (Figure 5). Water at the ice front (measured with the ALR and the ship CTD) is colder but lighter than water found deeper in the cavity (Figure 6). The temperature (Figures 6 and 5) and salinity (not shown) in the cavity generally increase with depth. The presence of warmer, saltier, and denser water in the cavity than at the ice front may indicate seasonal or interannual variability in the properties of the water at the ice front (as described by Kim et al. (2021)) and thus of water flowing into

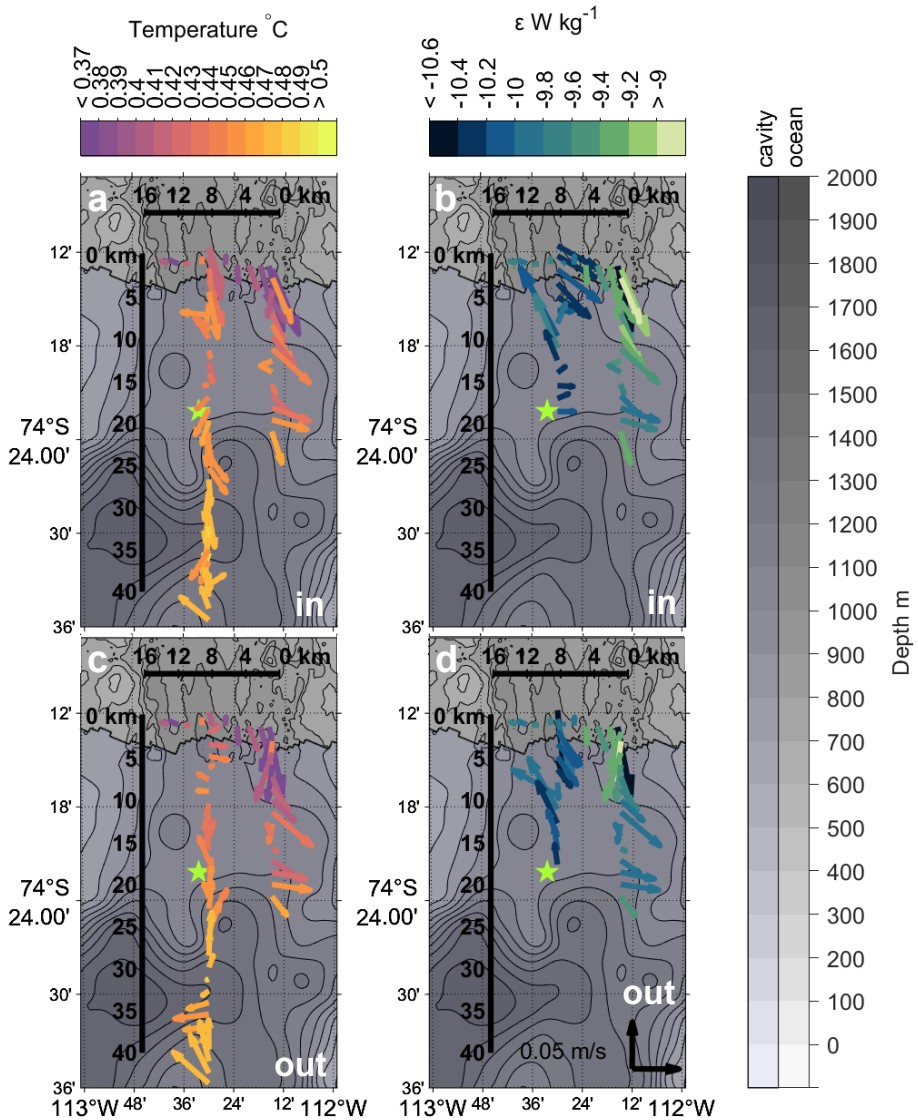

**Figure 5.** ALR dive tracks in front of and beneath Dotson Ice Shelf. 40-minute median depth averaged (median) currents along the dive tracks during the way into the cavity (a,b) and out of the cavity (c,d) coloured by conservative temperature (a,c) and turbulent kinetic energy dissipation rate (b,d). Star marks the location of a CTD profile obtained through a hole melted through the ice shelf into the cavity. Bathymetry as for Figure 2.

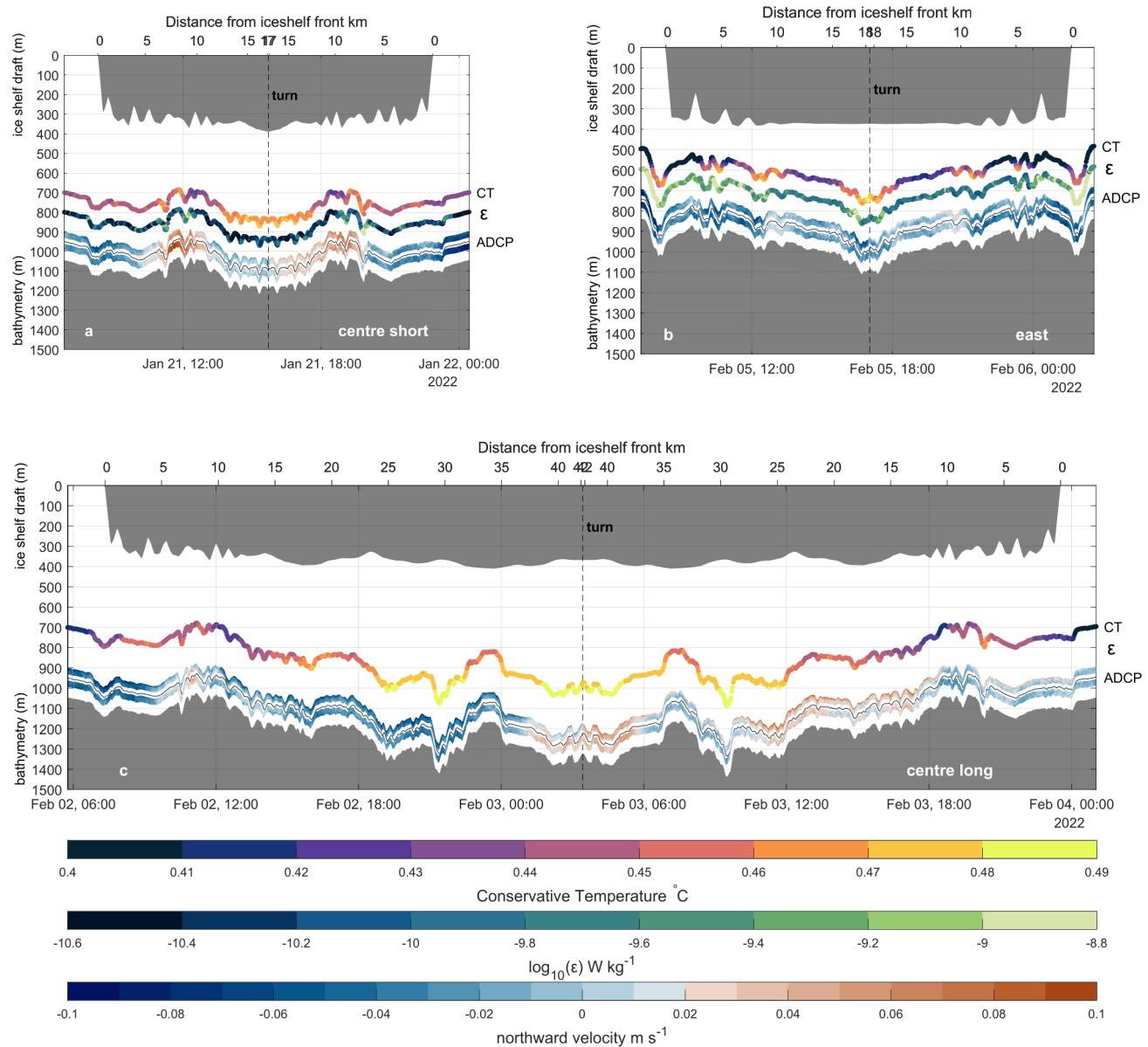

**Figure 6.** ALR dive tracks into Dotson ice shelf cavity for the centre_short (a), east (b) and centre_long (c) missions. The depth at which the ALR was flying is shown in the thin black line; the northward ADCP current velocity is shown above and below this line. We also show turbulent kinetic energy dissipation rate ($\varepsilon$) and Conservative Temperature (CT) measured by ALR, but offset vertically here for clarity. No turbulent kinetic energy dissipation values are available for the centre_long mission. The small labels to the right of the coloured lines indicate the variable measured, and corresponding colour bars are below panel c. The bathymetry is as measured by the ALR, and the depth of the iceshelf base is from Bedmachine V3 (Morlighem, 2022). Vertical dashed lines indicate where the ALR turned back.

the cavity. Alternatively, warmer water might be able to enter the DIS cavity from the neighbouring Crosson Ice Shelf cavity (indications of a deep connection are described in (Girton et al., 2019), however, they observed flow from DIS to Crosson).

### 3.2.2 Turbulent mixing in the cavity

Very few studies have successfully measured turbulent kinetic energy dissipation rate in ice shelf cavities (Kimura et al., 2016; Davis et al., 2022, 2025; Venables et al., 2014; Davis and Nicholls, 2019b), making our observations a valuable addition to our knowledge on cavity mixing. The ice–ocean interface at DIS shows evidence of a highly complex and spatially variable melt regime (Wåhlin et al., 2024a). There are indications that double diffusion and convection play a role in vertical heat and salt transport in this region of the cavity (Wåhlin et al., 2024a), and this should be investigated in future AUV microstructure missions. Our study did not target mixing and heat transport at the ice–ocean interface, or at the interface between the mCDW and the buoyant meltwater. Instead, we found evidence of a highly spatially variable pattern of turbulent kinetic energy dissipation rate close to the seabed under DIS. We find median rates of turbulent kinetic energy dissipation $\varepsilon$ of $10^{-11} - 10^{-10} \, \mathrm{W\,kg^{-1}}$ and median rates of diapycnal diffusivity $\kappa$ of $10^{-5} - 10^{-4} \, \mathrm{m^2\,s^{-1}}$. Maximum values were $10^{-7} \, \mathrm{W\,kg^{-1}}$ ($\varepsilon$) and $10^{-2} \, \mathrm{m^2\,s^{-1}}$ ($\kappa$), respectively (Table 2).

The highest levels of turbulent mixing occur in the inflow region at the ice front and in the east dive track, decreasing into the cavity (Figure 5 and Figure 6). The east dive track clearly shows the highest values for $\varepsilon$ of the three ALR dive tracks at DIS (Figure 8). The range, maximum and median values of $\varepsilon$ measured with the VMP at the ice front are higher than those observed in the cavity with the ALR, but ranges have a wide overlap. We compare our observations of turbulent kinetic energy dissipation rate with other observations under Ronne Ice Shelf (measured using a MicroRider mounted on an ALR; Davis et al., 2022), George VI Ice Shelf (measured with a VMP through a borehole; Venables et al., 2014), Thwaites Ice Shelf (measured with a VMP through a borehole; Davis et al., 2025) and Larsen C ice shelf (measured with a turbulence instrument cluster moored close to the ice–ocean interface; Davis and Nicholls, 2019b). The distributions of $\varepsilon$ under Ronne and George VI have similar shapes and ranges to our observations (Figure 8). The VMP observations do, however, show much higher maximum values. This is likely caused by the greater vertical extent of the VMP measurements, which reach into the ice–ocean boundary layer where $\varepsilon$ is elevated (Davis et al., 2025). This is confirmed by the measurements 2.5 m and 13.5 m from the ice-ocean interface under Larsen C, which show the highest average values of $\varepsilon$ of the measurements included in Figure 8. Further studies are needed to establish whether observed differences between ice shelves are driven by different mixing regimes or different observation techniques. The current state of knowledge leads us to conclude that the measurements taken under Dotson agree remarkably well with available distributions of $\varepsilon$ from other ice shelves, outside of the ice–ocean boundary layer.

Under DIS, areas of high turbulence coincide with regions of steep bathymetry and high along-slope current speed (Figures 9 and 7). These areas frequently exhibit positive temperature anomalies: the temperature in the turbulent patch is higher than the average temperature at that depth which indicates that warmer water is mixed upwards from below. They also coincide with areas of high vertical current shear and high along slope velocity (Figures 9 and 7). The relationship between high turbulent kinetic energy dissipation rate, high along-slope velocity, higher than average temperature and elevated current shear is most pronounced in the east dive track (Figure 9). The centre_short and along dive tracks (Figures 5 and 6) show much weaker

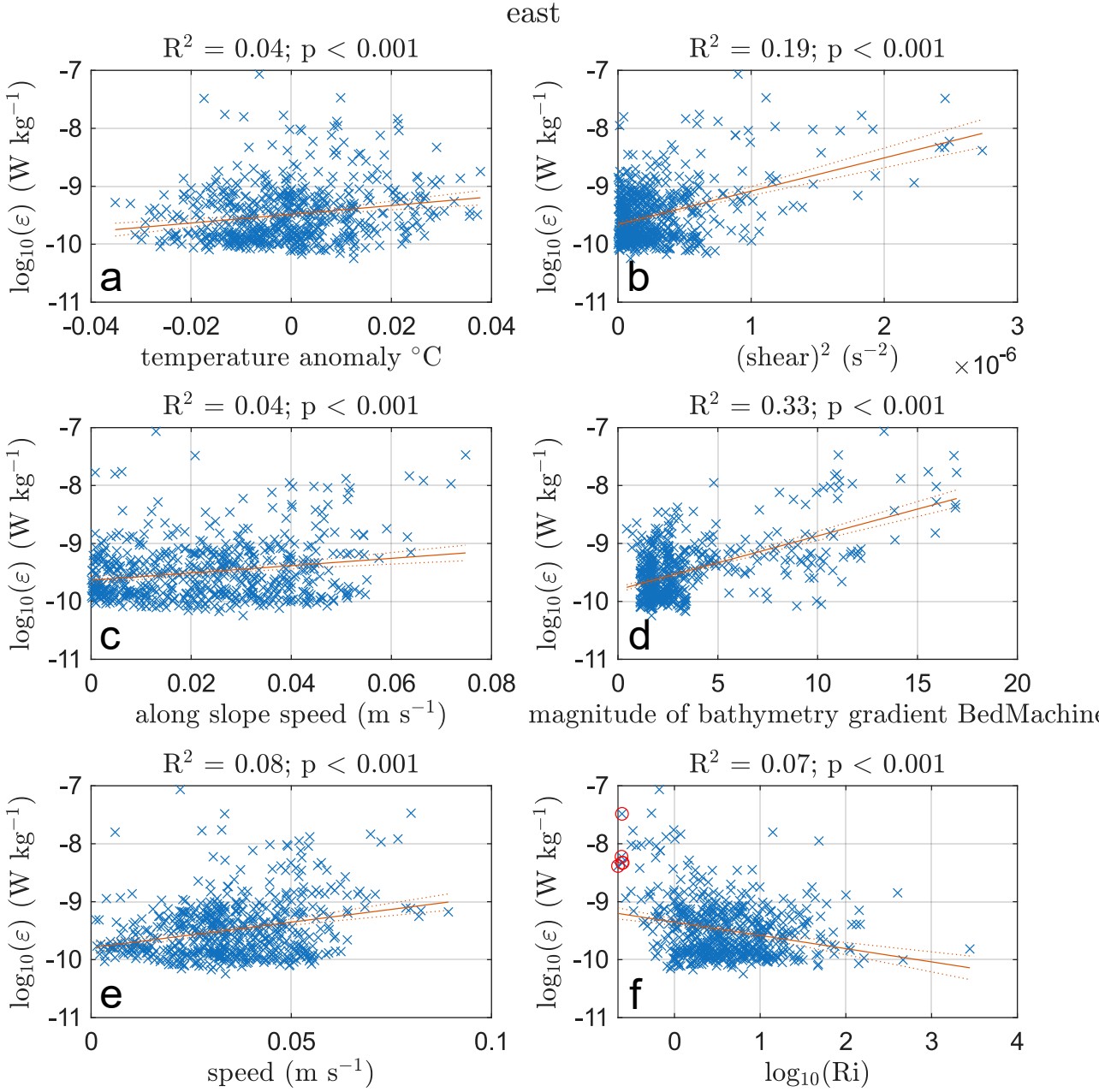

**Figure 7.** Turbulent kinetic energy dissipation rate plotted against drivers (blue crosses) with linear fit (solid red line) and 95 % confidence bounds for the fit (dotted red line). f: red circles mark where Ri < 0.25. $R^2$ is the percentage variance explained by the driver, p is the f-test percentage likelihood that the slope of the fit is zero.

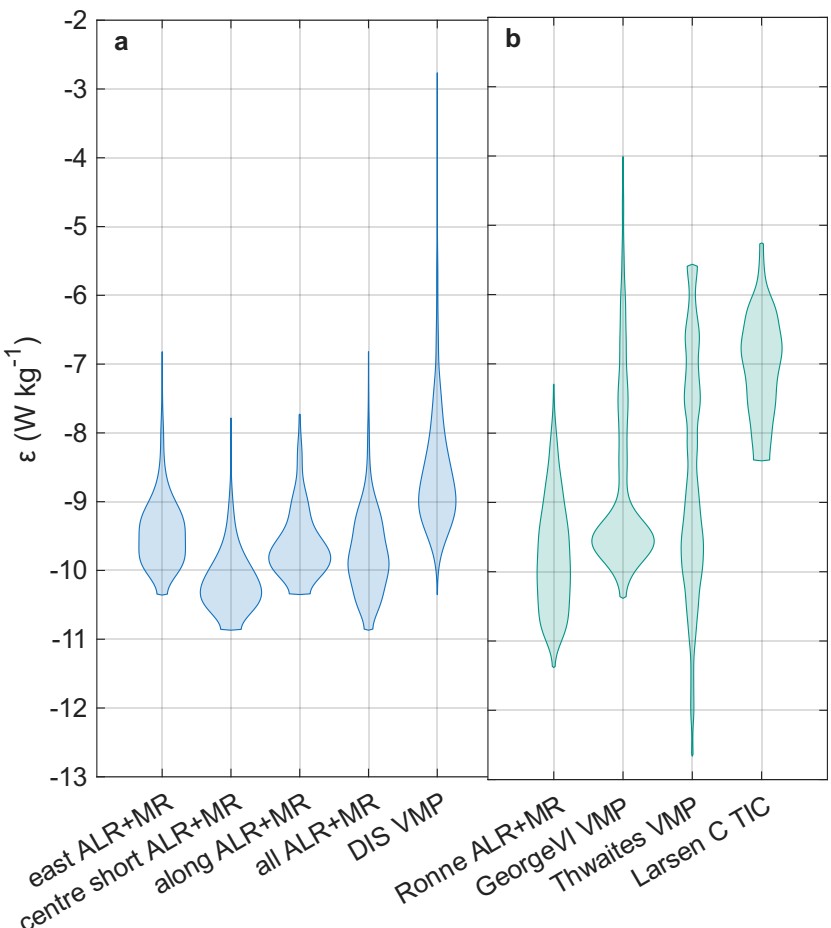

**Figure 8.** Kernel density distributions of turbulent kinetic energy dissipation rate from this study (a) and the literature (b). ALR: Autosub Long Range; MR: MicroRider; VMP: Vertical Microstructure Profiler. Names of dive tracks under Dotson as in Figure 2 and Table 1.

connections between these variables, possibly influenced by the overall lower turbulent kinetic energy dissipation rate and current speeds encountered at those locations. Ri is low in the area of high turbulent kinetic energy dissipation rate observed along the east dive track at $-1\,\mathrm{km}$ from the ice front (Figure 9). In areas of high turbulent kinetic energy dissipation rate, we find high absolute values of $q$ (positive and negative q occurs), and both symmetric and centrifugal instability (not shown). None of the variables with a statistically significant correlation with high turbulent kinetic energy dissipation rate (higher temperatures, high shear, high along-slope flow, etc.) are sufficient conditions for high turbulence (Figure 7). Additionally, no single parameter used to describe the potential for turbulence occurring (Ri, $q$, Ro, the stability criteria from Thomas et al. (2013)) is able to fully describe the pattern of turbulent kinetic energy dissipation rate we observe. Turbulence is patchy, episodic, and likely to be caused by a combination of factors.

The current direction relative to the bathymetric slope and the steepness of the slope are seen to be important among the parameters indicative of high turbulence (Figures 9 and 7). Regions of high turbulent kinetic energy dissipation rate coincide with areas of high along-slope current speed. We are limited in our resolution of the bathymetry under Dotson, with the BedMachine V3 bathymetry (Morlighem et al., 2019; Morlighem, 2022) clearly unable to resolve small scale features in the seabed (Figure 9). The effect of bathymetry on rates of turbulent kinetic energy dissipation is clearest in areas where the
bathymetry from BedMachine, used to calculate the bed's gradient, most closely matches the bathymetry measured by ALR (around $0\,\mathrm{km}$ on Figure 9). The bed and ice base of DIS show highly complex features related to ice shelf melt (this study and Wåhlin et al. (2024a)) which indicates the importance of combined multibeam and microstructure measurements, which should be attempted in the future. This high spatial variability and the effects of bathymetry under DIS are confirmed by the highly spatially variable turbulent kinetic energy dissipation rate found under two other ice shelves, Pine Island Ice Shelf (PIIS;
Kimura et al. (2016)) and Filchner Ronne Ice Shelf (FRIS; Davis et al. (2022)).

      Kimura et al. (2016) measured turbulent kinetic energy dissipation rate under PIIS in the southeastern Amundsen Sea, with their Autosub vehicle surveying with a saw-tooth pattern from the sea bed to the ice-ocean interface. In general, the PIIS cavity is very similar to the DIS cavity, both are warm cavity ice shelves with low tidal velocities and deep, rough beds. We observed similar levels of background mixing, with turbulent kinetic energy dissipation rate under DIS and PIIS between
$10^{-11} - 10^{-10}\,\mathrm{W\,kg^{-1}}$. Maximum rates of turbulent kinetic energy dissipation were also comparable, with values under PIIS and DIS on the order of $10^{-7}\,\mathrm{W\,kg^{-1}}$. However, the location of the ALR dive tracks and the dive patterns under PIIS and DIS were very different. We observed our highest mixing values in the bottom intensified inflow to the cavity, whereas Kimura et al. (2016) observed the highest levels of mixing close to the grounding line. Our ALR dive tracks did not reach the grounding line, and the dive tracks of Kimura et al. (2016) did not cover the inflow of the PIIS, making comparison difficult. Naveira
Garabato et al. (2017) did not find enhanced mixing in the PIIS inflow. Kimura et al. (2016) hypothesised that high (horizontal) density gradients driven by temperature differences and a bathymetric ridge can drive a baroclinic current with strong vertical current shear. This high shear in turn drives high levels of turbulence at the ridge under PIIS. Our study shows that high density gradients are not a requirement for high levels of turbulence. The ALR dive tracks under DIS are all located in mCDW with very low vertical density and temperature gradients ($< 0.008\,\mathrm{g\,kg^{-1}}$ per $100\,\mathrm{m}$ and $< 0.03\,°\mathrm{C}$ per $100\,\mathrm{m}$, respectively), but
we nevertheless record turbulence values of the same order of magnitude as Kimura et al. (2016). This shows that even in ice shelf cavities with similar far-field forcing, which experience warm CDW inflow over a deep rough bed, and show similar median and maximum rates of turbulent kinetic energy dissipation, the spatial distribution and drivers of the mixing can be very different. The forcing and mixing observed at the PIIS ridge (Kimura et al., 2016) may have an analogue in the DIS cavity at the bathymetric ridge close to the Kohler West grounding line (Figure 1). Future missions that penetrate deeper into the DIS
cavity are needed to describe the mCDW transport, mixing and melt regime at the DIS grounding line. This would allow a more detailed comparison with the PIIS cavity and the work by Kimura et al. (2016).

      Davis et al. (2022) measured turbulent kinetic energy dissipation rate under FRIS, a cold cavity ice shelf in the southern Weddell Sea. Their study followed a square saw-tooth pattern, switching between bottom and surface tracking while maintaining a distance of at least $80\,\mathrm{m}$ from the seabed and ice shelf. This pattern is an effective way to optimize turbulence measurements

while resolving the vertical structure in the cavity. FRIS experiences strong tidal forcing, the cavity has relatively low water-column thickness at the study site and the sea bed is virtually flat (Davis et al., 2022), whereas DIS experiences weak tidal forcing and the sea bed is rougher and deeper. Nonetheless Davis et al. (2022) recorded similar background mixing levels under FRIS as this study did under DIS, with average turbulent kinetic energy dissipation rates of $10^{-10}\,\mathrm{Wkg^{-1}}$ compared with our median values of $10^{-11} - 10^{-10}\,\mathrm{Wkg^{-1}}$ (Figure 8). Although we see high turbulent kinetic energy dissipation rates associated with high current speeds (Figure 9 at $0\,\mathrm{km}$, $5\,\mathrm{km}$ and $7\,\mathrm{km}$)), and although this relationship is statistically significant over the entire dive track (Figure 7), the relationship only explains $\sim 8\%$ of the variability in $\varepsilon$. Davis et al. (2022) do not find a statistically significant relationship between current speed and turbulent kinetic energy dissipation rate. Instead, Davis et al. (2022) found that turbulent kinetic energy dissipation rate is elevated to values of up to $10^{-8}\,\mathrm{Wkg^{-1}}$ in areas of high vertical current shear, which matches our observations. Under DIS, shear is significantly correlated with $\varepsilon$ and explains $\sim 19\%$ of variability in $\varepsilon$ (Figure 7) along the east dive track.

Davis et al. (2022) saw no increase in turbulence at the FRIS front, despite the current having to navigate a step in water column thickness induced by the ice shelf draft (Wåhlin et al., 2020). At DIS we see increased rates of turbulent kinetic energy dissipation close to the ice front in the east dive track (Figure 6b), but not in the centre_short dive track (Figure 6a). We argue that the bathymetric feature co-located with the ice-shelf front in the east dive track and absent in the centre_short track is more likely to be the driver of turbulent kinetic energy dissipation rates at the DIS front than the ice front draft. This is supported by the high correlation between the bathymetry gradient and $\varepsilon$ along the east dive track, with $\sim 33\%$ of the variability in $\varepsilon$ explained by the steepness of the gradient (Figure 7).

Turbulent mixing observations outside an ice shelf cavity in another embayment on the West Antarctic Peninsula, Ryder Bay, found turbulent kinetic energy dissipation rates on the order of $10^{-8}\,\mathrm{W\,kg^{-1}}$ above a bathymetric ridge (Scott et al., 2021), comparable to high turbulent kinetic energy dissipation rates we see above the bathymetric feature at $0\,\mathrm{km}$ in Figure 9. Enhanced mixing at ridges may be due to breaking of internal waves (e.g. Polzin et al., 1997), hydraulic control of currents flowing over steep bathymetry (e.g. Alford et al., 2013), or eddies in the wake of bathymetric obstacles (e.g. Muchowski et al., 2023). Our maximum turbulent kinetic energy dissipation rates are an order of magnitude higher than those observed under FRIS by Davis et al. (2022), even though Davis et al. (2022) report velocities almost twice as high as we see under DIS. This may be due to the rougher topography under DIS. While we see the topography vary by 10s of m per $100\,\mathrm{m}$, with troughs over $100\,\mathrm{m}$ deep, the depth of the bed in the area under FRIS observed by Davis et al. (2022) changes no more than approximately $10\,\mathrm{m}$ over the entire $22\,\mathrm{km}$ track. This study and the studies by Scott et al. (2021), Kimura et al. (2016), and Davis et al. (2022) show the need for repeat observations of turbulence over a wide variety of locations within ice shelf cavities. It additionally shows the importance of high resolution bathymetry within ice shelf cavities, with the currently available gridded products too coarse to resolve rough bathymetry that is a crucial driver of turbulent kinetic energy dissipation rates (Figure 9). Combined multibeam and microstructure observations would allow us to accurately understand and quantify the effect of rough bathymetry on flows within cavities.

## 3.3 Heat and salt fluxes in the cavity

Maximum and median values of diapycnal diffusivity $\kappa$, vertical heat flux $Q_T$, and vertical salt flux $Q_S$ from our observations under DIS are given in Table 2. Our median values of diapycnal diffusivity ($O(10^{-4}\,\text{m}^2\,\text{s}^{-1})$–$O(10^{-5}\,\text{m}^2\,\text{s}^{-1})$) are the same order of magnitude as globally-averaged ocean values (Waterhouse et al., 2014). The maximum values of diapycnal diffusivity in our study ($O(10^{-2}\,\text{m}^2\,\text{s}^{-1})$–$O(10^{-3}\,\text{m}^2\,\text{s}^{-1})$) match values observed close to the seabed over rough terrain or at ridges (Waterhouse et al., 2014).

Our observations under DIS provide valuable metrics against which turbulent mixing processes in numerical models could be assessed. Turbulent kinetic energy dissipation dissipation is not commonly modelled or parameterised in regional or global models. Instead, diapycnal diffusivity $\kappa$ is parametrised. A common parametrisation of diapycnal diffusivity in ice shelf cavities is the vertical profile method from Large et al. (1994) (e.g. in ROMS; Gwyther et al. (2015) or MITgcm; Nakayama et al. (2017)) which assumes higher values of $\kappa$ in boundary layers than in the interior. The interior mixing is made up of contributions from internal waves (parameterised as a constant), from shear instability (parameterised from the gradient Richardson number), and from double diffusion (parameterised from the double diffusion density ratio) (Large et al., 1994). The ice base roughness has been shown to influence the ice–ocean boundary layer mixing and the heat and salt flux into the boundary layer, and thus the spatial and temporal distribution of ice shelf melt (Gwyther et al., 2015). We are not aware of studies investigating the effects of spatially variable bottom boundary layer roughness on mixing and basal melt in an ice shelf cavity. The range of values for $\kappa$, the spatial variability, and forcing mechanisms we discuss, can be compared to the values and variability of the $\kappa$ profile parametrisation. This may allow a better understanding of the contribution of different drivers to mixing and of how realistic model mixing is.

Another common choice to parametrize mixing, used in the ISOMIP+ protocol (Asay-Davis et al., 2016), is to prescribe constant values for $\kappa$ in the vertical and horizontal, with higher values where the water column stratification is unstable. In stably stratified water, as under DIS, the ISOMIP+ protocol sets as $\kappa_{v,stable} = 5 \times 10^{-5} m^2 s^{-1}$ (Asay-Davis et al., 2016). The value of $\kappa$ used in ISOMIP+ has the same order of magnitude as the median value in the centre_short dive track ($2 \times 10^{-5} m^2 s^{-1}$), but is an order of magnitude lower than the median $\kappa$ on the east dive track ($1.1 \times 10^{-4} m^2 s^{-1}$) and 2–3 orders of magnitude lower than the maximum values we find within the cavity (Table 2). Thus, the constant value of $\kappa$ used in ISOMIP+ is a good choice for slow flows with low shear over smooth topography, but may underestimate mixing in other areas which may in turn influence modelled ice-shelf melt.

Our median vertical heat fluxes are positive (upwards) and range between $0.11\,\text{W}\,\text{m}^{-2}$ and $0.02\,\text{W}\,\text{m}^{-2}$. This is the same order of magnitude as the median vertical heat flux measured in the mCDW under FRIS ($0.2\,\text{W}\,\text{m}^2$; Davis et al. (2022)). However, due to opposing temperature gradients, our heat flux is positive (upwards), whereas the heatflux at the interface between modified Warm Deep Water and High Salinity Shelf Water under FRIS is negative (downwards). From our calculations of maximum and median temperature ($Q_T$) and salt fluxes ($Q_S$) for the centre_short and east dive tracks (Table 2), we estimate the heat that might be lost through upward vertical mixing as the warm inflow travels from the ice front to the grounding line. We assume a distance of $80\,\text{km}$ from the ice front to the grounding line of DIS, consistent with the distances traveled

by neutrally buoyant floats following the warm inflow into the DIS cavity (Girton et al., 2019). Our mean meridional velocity in the cavity is $-0.01 \, \mathrm{m \, s^{-1}}$, which results in a travel time of 92 days. This is longer than the travel times of $1.5 - 2$ months estimated from inflow velocities measured at moorings in front of DIS (Milillo et al., 2022; Yang et al., 2022) and from floats released into the DIS cavity (Girton et al., 2019). Since our ALR dive tracks do not lie within the core of the mCDW inflow, our southward velocities are likely an underestimate. Thus, we use a travel time of 2 months, as suggested by Milillo et al. (2022) and Girton et al. (2019) for our calculations. Using the mean of the heat fluxes in Table 2 and Equation 4, we calculate the temperature decrease of the bottom $100 \, \mathrm{m}$ of the water column in the cavity during its passage to the grounding line to be $8 \times 10^{-4} \, \mathrm{°C}$. It would take 200 years for the temperature of the $100 \, \mathrm{m}$ thick bottom layer in the cavity to decrease by $1 \, \mathrm{°C}$. This calculation demonstrates that the levels of turbulent mixing experienced by the warm mCDW inflow to the DIS will lead to negligible loss of heat during its path to the grounding line, leaving plenty of heat available to melt the ice shelf base there.

We can estimate the DIS basal melt, assuming that the entire heat flux is used to melt ice at a depth of approximately $1000 \, \mathrm{m}$. With this assumption the melt rate $m$ is

$$m = \frac{Q_t}{L_i \rho_i} \tag{10}$$

with $L_i = 3.315 \times 10^5 \, \mathrm{J \, kg^{-1}}$ the latent heat of fusion at $1000 \, \mathrm{dbar}$, and $\rho_i = 917 \, \mathrm{kg \, m^{-3}}$ the density of ice. This results in melt rate estimates of $2 \times 10^{-3} \, \mathrm{m \, yr^{-1}}$ to $11 \times 10^{-3} \, \mathrm{m \, yr^{-1}}$. Published estimates for area averaged melt rates under DIS range from $6.1 \, \mathrm{m \, yr^{-1}}$ to $18.3 \, \mathrm{m \, yr^{-1}}$ (Gourmelen et al., 2017; Lilien et al., 2018; Robertson, 2013; Jenkins et al., 2018; Schodlok et al., 2012) with some estimates up to $32.9 \, \mathrm{m \, yr^{-1}}$ (Jenkins et al., 2018). The low upward heat flux within the mCDW layer is thus not able to maintain the observed melt rates under DIS. To achieve the melt rate estimates from (Gourmelen et al., 2017; Lilien et al., 2018; Robertson, 2013; Schodlok et al., 2012; Jenkins et al., 2018) the vertical heat flux would need to he greater than $59 \, \mathrm{W \, m^{-2}}$ to $316 \, \mathrm{W \, m^{-2}}$, values three to four orders of magnitude larger than our median estimates and up to six times our maximum estimate (Table 2).

Davis et al. (2025); Davis and Nicholls (2019b) showed elevated levels of $\varepsilon$ in the ice–ocean boundary layer under Thwaites and Larsen C, and Kimura et al. (2016) observed elevated values of $\varepsilon$ close to the ice–ocean interface and over a bathymetric ridge in front of the PIIS grounding line. In these areas high turbulent kinetic energy dissipation rate and high vertical and horizontal temperature gradients lead to high heat fluxes. Our study did not reach the ice–ocean boundary layer or the ridge limiting flow to the DIS grounding line (Figure 1) which may explain the underestimate of the area averaged ice shelf melt rate using the observed heat fluxes. The value for $\kappa$ for stably stratified water used in the ISOMIP+ protocol, matches our estimate of $\kappa$. Thus, modelled vertical heat transport, in regions for which this estimate is used, could also be too low to explain observed ice shelf melt rates. The low heat fluxes in the interior of ice shelf cavities would need to be offset by higher heat fluxes at the grounding line and in the ice–ocean boundary layer.

We can additionally estimate a melt rate for DIS from the temperature difference between the inflow and the outflow of the cavity and the average residence time of water within the cavity. We take the heat needed to warm the ice shelf to the

freezing point temperature and the heat needed to warm the melt water to the temperature of the outflow into account. A back-of-the-envelope calculation for melt rate gives:

$$m = \frac{V_{in} C_p \bar{\rho} (\theta_{in} - \theta_{out})}{\rho_i A_{DIS} (C_i (\theta_f - \theta_{ice}) + L_i + C_p (\theta_{out} - \theta_f))} \tag{11}$$

with $V_{in} = v_{cavity} A_{inflow}$, the volume transport in the inflow; $v_{cavity} = \frac{D}{t}$ the velocity of the inflow; $D$ the distance water has to travel from the ice front to the grounding line and back; $t$ the time the water takes to travel to the grounding line and back; $A_{inflow}$ the area through which water flows into the cavity; $A_{DIS}$ the area of the DIS; $C_i$ the specific heat capacity of ice at $-2\,°\mathrm{C}$ and $1000\,\mathrm{db}$; $\theta_f$ the freezing point temperature of seawater; $\theta_{in}$ the average temperature of the inflow to DIS; $\theta_{out}$ the average temperature of the outflow from DIS; $\theta_{ice}$ the far-field internal temperature of the DIS.

We assume the following values for these parameters: $D = 240\,\mathrm{km}$ (Figure 1); $t = 2\,\mathrm{months}$ (Milillo et al., 2022; Yang et al., 2022; Girton et al., 2019); $A_{inflow} = 500\,\mathrm{m} \times 15\,\mathrm{km}$ (solid box in Figure 3); $A_{DIS} = 5200\,\mathrm{km}^2$ (Lilien et al., 2018); $\theta_f \approx -2\,°\mathrm{C}$; $\theta_{in} = 0.2\,°\mathrm{C}$ (the average temperature in the solid box in Figure 3); $\theta_{out} = 0.17\,°\mathrm{C}$ (the average temperature in the dashed box in Figure 3, the outflow extends to shallower depths in the water column than the inflow due to the thinner ice shelf draft in the west of the DIS (e.g. Wåhlin et al., 2024a)); $\theta_{ice} = -25\,°\mathrm{C}$, an estimate of the far-field ice temperature. Our estimate of melt rate and heat flux from inflow and outflow temperatures is most sensitive to the area over which we average outflow temperatures (Figure 3) and represents an order of magnitude estimate only.

Equation 11 results in an estimate of the melt rate of $\sim 10 \pm 5\,\mathrm{m\,yr^{-1}}$, which lies within the range of published values (e.g. Gourmelen et al., 2017; Lilien et al., 2018; Robertson, 2013; Schodlok et al., 2012; Jenkins et al., 2018). To maintain this melt rate the vertical heat flux in the cavity would need to be $100 \pm 50\,\mathrm{W\,m^{-2}}$, about three orders of magnitude higher than the median values along the east dive track (Table 2). Rearranging Equation 11 allows us to estimate the percentage of the heat entering the ice shelf cavity that is used to melt ice. We estimate that the inflow transports $4 \pm 2\,\mathrm{TW}$ into the cavity and the melt takes up $0.6 \pm 0.4\,\mathrm{TW}$, thus, only $\sim 15 \pm 9\,\%$ of the heat entering DIS is used to melt the ice shelf. Modelling studies have estimated this value to be smaller, at 8 % (Jourdain et al., 2017), but within our error range. Transport calculations by Jenkins et al. (2018) yield the same range for heat flux into the cavity as our estimate does, however, their calculated melt rate, derived from melt water fluxes, is significantly higher ($6\,\mathrm{m\,yr^{-1}} - 33\,\mathrm{m\,yr^{-1}}$). These melt rates would require heat fluxes of $60\,\mathrm{W\,m^{-2}} - 317\,\mathrm{W\,m^{-2}}$. We need significantly more measurements under ice shelves to understand the role of mixing in different areas and regimes, and its effect on ice shelf melt rate.

Our study and that of Dotto et al. (2025) represent summer snapshots of ocean conditions in front of and underneath DIS. Due to the suggested strong seasonality of the inflow and outflow speed, heat and meltwater content at DIS (Yang et al., 2022), and the observation that turbulent kinetic energy dissipation rate is high where current speeds are high (Figure 9), our estimates of mixing and heat transport close to the bed of DIS may represent upper limits. To test this hypothesis, highly challenging wintertime observations of turbulent mixing, ocean velocity and ocean properties within ice shelf cavities are needed. The mismatch between heat fluxes observed by ALR under DIS and the heat fluxes necessary to maintain the observed basal melt rate under DIS, means that questions of seasonality, spatial variability and the effect of ridges under DIS need to be addressed

**Table 2.** Maximum and median values for key quantities along the east and centre_short dive tracks into the Dotson ice shelf cavity. Values are calculated from observations at their highest resolution (1 s for current speed, 32 s for all other variables), without prior smoothing or binning.

| | | east | | centre_short | |
| --- | --- | --- | --- | --- | --- |
| | | max | median | max | median |
| Current speed | $\mathrm{ms}^{-1}$ | 0.1 | 0.03 | 0.11 | 0.04 |
| $\varepsilon$ | $\mathrm{W\,kg}^{-1}$ | $1.5 \times 10^{-7}$ | $3.3 \times 10^{-10}$ | $1.6 \times 10^{-8}$ | $5.9 \times 10^{-11}$ |
| $\kappa$ | $\mathrm{m^2s}^{-1}$ | 0.05 | $1.1 \times 10^{-4}$ | $5.5 \times 10^{-3}$ | $2 \times 10^{-5}$ |
| Qt | $\mathrm{W\,m}^{-2}$ | 52 | 0.11 | 5.6 | 0.02 |
| Qs | $\mathrm{kg\,m^{-2}s}^{-1}$ | $5.1 \times 10^{-3}$ | $1.1 \times 10^{-5}$ | $5.5 \times 10^{-4}$ | $2 \times 10^{-6}$ |

in future missions. ALR has the capability to remain moored at the seabed for months, periodically waking up to perform missions before lying dormant again. Such a campaign, though risky, using an AUV or ocean gliders, should be considered in order to resolve seasonal variability in turbulent mixing under ice shelves such as DIS. The observations by Kimura et al. (2016) demonstrate that campaigns which resole the full water column thickness and extend to the grounding line are possible and should be attempted under other ice shelves.

## 4 Conclusions

We have presented the first measurements of the current and turbulence regime near the seabed under the Dotson Ice Shelf. We show that turbulent kinetic energy dissipation is highly spatially variable, indicating that further effort is needed to observe, model and classify bed roughness, stratification, heat content and turbulent mixing, and their effects on melting of the ice shelf base. Background turbulent kinetic energy dissipation was $10^{-10}\,\mathrm{W\,kg}^{-1}$. Higher turbulent kinetic energy dissipation ($10^{-7}\,\mathrm{W\,kg}^{-1}$) coincides with the mCDW inflow, regions of rough bathymetry, higher along slope current speed, high vertical current shear and high temperature anomalies. However, none of these drivers alone form a sufficient indicator of high turbulent kinetic energy dissipation rate. Our background $10^{-10}\,\mathrm{W\,kg}^{-1}$ and maximum $10^{-7}\,\mathrm{W\,kg}^{-1}$ rates of turbulent kinetic energy dissipation are comparable to those measured by previous surveys under ice shelves (Kimura et al., 2016; Davis et al., 2022, 2025; Venables et al., 2014; Davis and Nicholls, 2019b). Due to differences between the cavities studied and the observing techniques within the cavities, all five studies are able to resolve different mixing features, with the present study focusing on turbulent mixing over rough topography. We show that there are patches of elevated vertical heat flux distributed throughout the cavity, showcasing a mechanism for transporting heat from deep warm layers in the cavity toward the ice shelf base. Median values of vertical heat flux from turbulent mixing are low, showing that the mCDW in the cavity loses negligible heat on its way to the grounding line leaving plenty of heat available to melt the ice shelf base there. Estimates of ice shelf melt rate for DIS show that the low vertical heat flux in the bottom layer of mCDW are approximately three orders of magnitude too low to explain observed levels of ice shelf basal melt. Estimates of basal melt rate from DIS inflow and outflow temperatures

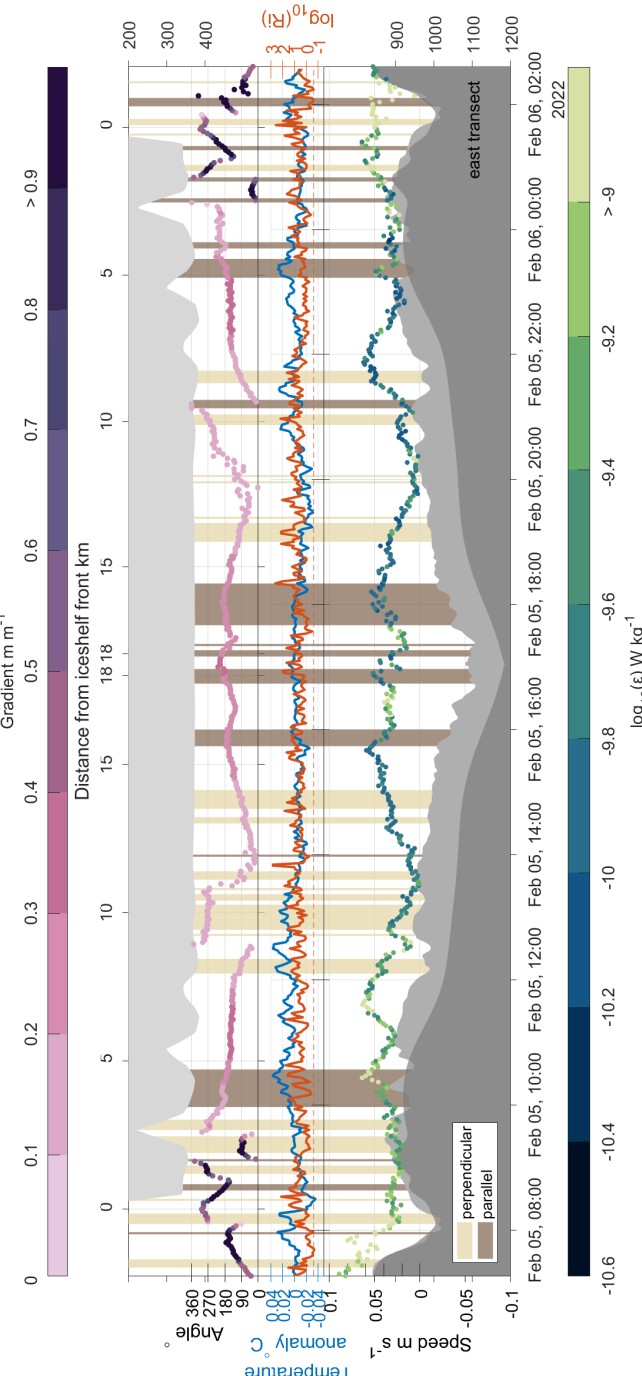

**Figure 9.** Timeseries of the east dive track into the Dotson Ice Shelf cavity. Top panel shows the angle at which the current direction intersects with the maximum bathymetric gradient (an angle of 0° is flow parallel to the isobaths with high ground to the right of the flow; an angle of 90° is downhill flow). Middle panel shows temperature anomaly (relative to the depth mean) and current shear. Bottom panel shows current speed coloured by turbulent kinetic energy dissipation rate, $\varepsilon$. Bathymetry measured by ALR is shown in light grey, and the smoothed bathymetry used to calculate the bathymetric gradient is shown in dark grey. Beige patches show where the current is perpendicular to the isobaths, and brown patches show where the current is parallel to the isobaths. The distance travelled by the ALR relative to the start of the dive track is shown on the top panel, time is shown on the bottom panel.

agree with published ranges of ice shelf melt rates and demonstrate that only a small fraction of the available heat in DIS is used to melt the ice shelf under current conditions.

*Data availability.* CTD and LADCP data along the ice front are archived at https://doi.org/10.15784/601785 (NSF/NERC ARTEMIS and ITGC TARSAN, 2024) and https://doi.org/10.5285/18a8be08-07c6-d76c-e063-7086abc01604 (Dotto, Tiago S et al., 2024), respectively. The CTD downcast through the DIS is archived at https://doi.org/10.5878/JEJ3-KV87. The turbulence data from under Larsen C ice shelf are archived at https://doi.org/10.5285/16ee2665-d0d0-41b9-a046-23b0a7369c61 (Davis and Nicholls, 2019a). The turbulence data from Ronne ice shelf are archived at https://doi.org/10.5285/eb2f66fa-1c64-49af-b9e8-ce3124ce3c03 (Davis and Jenkins, 2022). VMP data from Thwaites ice shelf are archived at https://doi.org/10.5285/2b33895b-5069-4c49-95bd-2624c980498b (Davis et al., 2024). All other data used in this study is archived at https://doi.org/10.5281/zenodo.15280916.

*Author contributions.* MER analysed the data, produced the figures, investigated the results, and wrote the paper. PEDD provided processed turbulence data from previous publications and advised on data processing. KJH and RAH acquired funding, discussed the results, and provided supervision. KJH and MER revised and edited the paper.

*Competing interests.* KJH is one of the co-editors-in-chief of Ocean Science.

*Acknowledgements.* This work is from the Thwaites-Amundsen Regional Survey and Network Integrating Atmosphere-Ice-Ocean Processes (TARSAN) project, a component of the International Thwaites Glacier Collaboration (ITGC). Support from National Science Foundation (NSF: Grant 1929991) and Natural Environment Research Council (NERC: Grant NE/S006419/1 and NE/S006591/1). Logistics provided by NSF-U.S. Antarctic Program and NERC-British Antarctic Survey. MER, KJH, and RAH were supported by TARSAN project Grant NE/S006419/1, KJH, and RAH were supported by ARTEMIS project (NE/W007045/1). PEDD was funded by the MELT project, a component of the International Thwaites Glacier Collaboration. Support from National Science Foundation (NSF: Grant 1739003) and Natural Environment Research Council (NERC: Grant NE/ S006656/1). Logistics provided by NSF-U.S. Antarctic Program and NERC-British Antarctic Survey. We thank Tiago S. Dotto and Alberto Naveira Garabato for discussions on the VMP sections, Eleanor Frajka-Williams for the use of her ADCP processing code. We thank Emily Venables for providing processed VMP data from under George Vi ice shelf. This is ITGC Contribution No. ITGC-145.

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
