# Peer review of "Observations of turbulent mixing in the Dotson Ice Shelf cavity"

_EGUsphere, 2025_

## Author Comment (AC1)

**Response to reviewer R1 comments**

June 23, 2025

Thank you for your detailed review, we will be commenting on the comments by reviewers R1 and R2 separately and upload full responses to reviewers, tracked changes and a revised manuscript after the discussion phase has ended. In this document, reviewer comments are in **black** and our comments are in **red**.

This is a clearly written paper with nice figures describing nice analysis of an extraordinarily rare and hard to obtain dataset. The manuscript should be published.

Thank you for your positive assessment of our paper, we will address your indiviual points below

I do have a number of comments, questions and morsels for thought that I list below in the order in which I read. The majority are (very) minor, amounting to text and grammar nits, but some are more substantive. In particular I would like to see

- more supporting evidence behind the claim that mixing is weak (for the reasons given in the final comment below),

  Could you elaborate on the type of evidence you are looking for? We show that the median TKE dissipation rate is $10^{-11}$ to $10^{-10}$, which are very low values, comparable to the background TKE dissipation rate

in the ocean.

- better figure 3 and 4, which currently mixes aspect ratios, has the reader going back and forth and does not allow direct comparisons of the most relevant quantities - specifically epsilon and the different instability indicators

We will adjust Figures 3 and 4 to have the same aspect ratio and trial different combinations of the data in our revised submission.

- quantification of the ADCP vertical wavenumber response and hence justification of the numerical values of Ri presented (or alternately toning down the reference to specific values such as $Ri = 1/4$ given the estimates are noisy and not fully resolved),

We are unsure what the reviewer is asking about here. Do you refer to the vertical wave number response that Polzin et al. (2002) refer to when estimating turbulent mixing processes from vertical shear in the ADCP? We do not use the ADCP to calculate mixing, we only use it to get information on the horizontal velocity in the vicinity of our microstructure shear measurements. The Richardson number is calculated from the vertical shear between successive 8 m tall (in the vertical) ADCP bins, but this is not used for the turbulent shear calculations. Could you clarify what you are concerned about here?

- justification for use of median versus mean

We use the median as it is less impacted by outliers or non-normal distribution of values. If the data is normally distributed the median and mean are identical, so there is no negative effect of using the median as the default method of averaging values.

- and finally and perhaps most substantively, an explanation for why the turbulent heat fluxes just above the bottom are important to measure. Ie, is that the water that will eventually meet the grounding line, or should the study have been done nearer the top of the mCDW watermass where the gradients and heat losses are much stronger?

*We will add information to the text to clarify that we measure heat fluxes close to the bottom to capture the effect of topography roughness on the flow, to capture the mixing where the bottom intensified warm inflow interacts with the seabed and due to practical constraints (the ALR needs to stay within 100 m of the seabed to allow for accurate dead-reckoning and bottom tracking).*

Good luck. I enjoyed reading the paper and hope that these comments are useful.

11: topography, turbulent or both not resolved?

*We will clarify this sentence to confer that turbulent mixing is not resolved in models and topography is not resolved in bathymetry products or models.*

26: awkward

35, 53: "this" is a weak reference. Please reword; see Strunk and White if needed.

48: Melt rates two words?

*We will correct this*

52: This statement is actually not true: epsilon is the dissipation rate and further assumptions must be invoked to infer the mixing. This needs to be corrected and expanded upon.

*We will expand our wording here to emphasis that certain assumptions need to be fulfilled to associate epsilon with turbulence, such as Taylor's frozen turbulence hypothesis.*

55: This would be a good place to distinguish what is different about this study from the other two.

*We will add words to that effect here.*

78 56: which $->$ that. Also, is this the only reason mixing is important to
79 know for these situations?

80 mixing at the seabed – ocean interface is also important for nutrient trans-
81 port, such as the transport of iron from sedimentary sources to the euphotic
82 zone. We refer to such processes in the paragraph above.

83 66-68: Please give order of magnitude of the clock offsets before correction
84 and the precision of the alignment afterwards.

85 We will add this information in the revised manuscript.

86 74: Please explain why you used median instead of mean?

87 See our explanation above.

88 95: Could indicate this is likely because of $F = ma$; ie the same force on the
89 huge autos produces much smaller accelerations.

90 We will add a note in the revised document speculating that the lower impact
91 of vibrations on ALR compared to smaller craft is due to the greater weight
92 of the vehicle.

93 105: on which this study focuses.

94 This will be changed in the revised manuscript.

95 105 general: is this the first paper that presents the details of shear mi-
96 crostructure from Autosub? Surprising if so but if true, you might consider
97 showing a few spectra and additional details, possibly in an appendix, so
98 that future work can cite this paper.

99 This is not the first such paper, we refer the reader to Davis et al. (2022)
100 for information of the spectral response of the shear probes on ALR. This
101 reference will be added to the revised manuscript.

102 111: Shih et al is a very bad reference for this! They find a $Re_b$-dependent
103 Gamma. Suggest just citing Osborn (1980). There are also now a handful of

observational references supporting the assertion that $\Gamma = 0.2$.

Thank you for ppointing this out, we will remove the reference to Shih. If you can point us toward the observational studies that find $\Gamma = 0.2$, we would be interested in seeing them.

113: How close to the bottom of the ice is the shallowest CTD measurement shown? The very strong gradients at the very top of the cavity CTD casts (Fig 2 black) are interesting.

The CTD cast goes right to the ice – ocean interface. We refer the reader to Wåhlin et al. (2024) for a discussion of the CTD measurements at the interface.

123 and throughout: I believe units should be in roman, not italicized, font.

We have corrected this where we found such instances, all remaining formatting will be finalized in the copy editing process.

136: Suggest reformatting the equation.

This will be reformatted in the revised document

140: Please make it very clear that Ri (under the ice at least) is based on a single N2 profile whereas the shear is a function of location and time. This is OK, but appropriate caveats as to its governing local instabilities without in-situ N2 should be given.

We will add additional words to this effect to the revised manuscript

173: Generally, avoid "there is" in favor of more active language such as "flow is to the . . . "

We will change some of our wording where we deem appropriate in the revised manuscript

177: High compared to what?

We will add context that epsilon is high compared to the range of epsilon

observed at the ice shelf front.

177: runon sentence.

In addition to adding context (see above), this sentence will be split into shorter sentences.

Figure 3, lines 2 and 4 of caption: runon sentences. Also, the dots are said to indicate the starting locations - but they are a continuous line. I'd have thought there would just be two starting locations, one for center and one for east? Please clarify.

We are unsure what you mean by "dots are said to indicate starting locations". There are two large dots with a black outline that show the values at the starting locations of the east and centre tracks. All other dots show the 10-minute median of the values of the "along" dive track. If we make it clear that we plot the 10-minute median of the ALR values along the dive track, would that solve the issue?

Figure 4: Personally I think it would be better to keep the aspect ratio constant between Fig 3 and 4. Also, sine you already plotted velocity in Figure 3, suggest including a panel of N2. The aspect ratio is all the more a problem later when the authors are comparing epsilon to the different instability indicators - but the reader must go back and forth between figure 3 and 4. Suggest standardizing the aspect ratio and including an epsilon panel in Figure 4. Possibly even adding Ri contours to the epsilon panel or epsilon contours to the Ri panel since the authors are trying to demonstrate correspondence between the two quantities.

Thank you for this feedback, we will standardise the aspect ratio between Fig 3 and Fig 4 and trial your suggestions regarding N2, epsilon and Ri in the revised manuscript. However we may decide not to adopt a specific suggestions due to readability or an excessive number of subpanels.

Also, the Ri panel is just a big sea of red. Consider plotting something else to highlight the unstable regions such as $Ri^{-1}$ or $Fr = Uz/N$.

The Ri panel is mainly red due to the choice of colourbar. We chose to plot $Ri < 1/4$, $1 > Ri > 1/4$, and $Ri > 1$ as three different colours in keeping with established practice to distinguish along criteria for instability. Plotting $1/Ri$ would make it less obvious where $Ri < 1/4$. We would like to avoid plotting additional instability metrics such as the Froude number to avoid confusion.

182: Doesn't negative PV mean unstable? The whole water column is unstable? Is it backwards in the southern hemisphere? Some statements to clarify would be useful.

Instabilities may develop when PV and f have opposite signs, as f is negative in the southern hemisphere, PV > 0 indicates conditions favourable to instability. We will clarify this in the text.

188: I don't agree with this statement - the high dissipation does not appear to me to line up at all with for Ri. Furthermore, given the ADCP's finite vertical resolution and noise, some additional detail needs to be given on how seriously we are to take the numerical value of Ri. I think that either some wavenumber spectra and transfer functions a la Polzin 2002 need to be included, or Ri used as a qualitative indicator.

As far as we understand Polzin et al., 2002 the vertical waenumber response of the ADCP is relevant when calculating turbulent dissipation from the ADCP. We are not using the ADCP for turbulence. We use the VMP or microrider for shear microstructure and the LADCP and ADCP on the ALR to get an idea of the vertical and horizontal structure of the water column at much larger scales, a background value if you will. Ri is frequently calculated from LADCP output with bin sizes of 8 m. If you have additional concerns could you be more specific on how Ri would be affected by the ADCP resolution? And what you mean by "wavenumber spectra and transfer functions"?

191: I disagree; elevated mixing is much broader than the regions of $Ri < 1/4$ - augmenting my previous point.

We will clarify that the high epsilon includes, but extends beyond, the region of low Ri. Can you clarify how this is related to your previous point?

193: This statement is not justified. Epsilon appears surface intensified as well. And while it is bottom intensified, I do not think the statement that it is heightened over rough topography, shear or high currents (of which you generally must choose either high current or high shear, not both...) is supported. And as before, I don't think that high epsilon lines up with low Ri either. Either way, if this statement is retained, more analysis needs to be shown - scatter plots, binned averages, etc.

We will clarify that we are only considering epsilon below the Winter Water layer (approx 400 m), thus we do not discuss high epsilon at the surface. Can you clarify why you think high current speed can not coincide with high current shear? We will include scatter plots in our response to reviewers, however, as we state in the manuscript, the relationship between topography, shear and epsilon that we find around 0 km in the east dive track is not valid everywhere. Additionally, the bathymetry gradient deeper in the cavity is affected by the low resolution of bedmachine, making correlations and scatter plots noisy.

197: runon sentence. And seemingly unrelated sentences. Ri governs shear instability, not symmetric instability... (I understand they are highly correlated here, but they are different, so clarification is needed).

We will insert a paragraph break before "at the nearby Pine Island Ice Shelf....". We will clarify that SI is not governed by Ri.

202: What is a barotropical jump?

It is an oceanographic term for an abrupt change in water column thickness. This occurs at the ice shelf front, since ocean currents want to flow along lines of uniform water column thickness, the ice shelf draft poses a barrier to flow, even at depths deeper than its draft.

207: Please rewrite this passive and vague sentence.

We will rewrite this sentence in the revised manuscript.

204-210: Suggest moving this speculative bit to the discussion.

We originally had results and discussion split, but chose to integrate them to avoid duplicating information and to limit jumping back and forth between topics. We will retain this structure.

216: I think it would be nice to compare this to open ocean values at a similar depth and/or abyssal values, for context. Otherwise "weakly stable" doesn't have meaning.

We will add typical open ocean values for N2.

218: Style guides such as Strunk and White suggest avoiding "Figure x shows..." in favor of "statement x is true (Figure y)."

We will rephrase this sentence.

223: Figure 6 and 5 -> Figures 5 and 6

Thank you for pointing out this typo

236: redundant. Suggest "Maximum values were" or "Values reached."

We will change this in the revised manuscript

238: Again, I'm afraid I don't see this. There are counter examples where epsilon is high over flat bottoms. Please include plots that allow direct comparison such as plotting epsilon with Ri, current speed or bathymetric slope over plotted, or scatter plots or binned averages (e.g. epsilon(Ri) etc) if you want to make this claim.

We will clarify that we are not making statements about a universal relationship. Turbulence is influences by many factors, some only incompletely resolved in our study, or not resolved at all. Our statement is a qualitative observation that can explain many of the observed patches of high turbulence, though you are right that there are regions where this relationship

may be absent or obscured by the limitations of our study.

241: Please remind reader that it's Ri computed from in-situ where and

The remainder of this sentence seems to be missing, please clarify

245: Again, please include transfer function and instrument response information if you wish to quantify the numerical value of Ri versus using it as a qualitative indication. Note as well that these transfer functions and hence the mapping of true to measured Ri will be different for the Autosub and the LADCP.

Can you clarify what you mean by the mapping of true to measured Ri and how that is influenced by the ADCP? It is common practice to calculate Ri from vertical shear from the 8 m binned ADCP and we do not use the ADCP to calculate fine scale microstructure.

257: Is it really necessary to use a package like this to compute a spatial gradient? More fundamentally I do not see a relationship between RMS bathymetric slope and dissipation rate.

I do not use a package to calculate the gradient. The bathymetry from ALR is only 1D, to get a 2D gradient I use bedmachine to get the bathymetry normal to and along the ALR dive track. Can you clarify what you mean by "RMS bathymetric slope"? We do not calculate RMS of the slope and to our eyes there is a clear relationship between the bathymetry and epsilon close to 0 km on the east dive track.

264 onwards: consider moving all of this comparison to past work to the discussion, so that the results section just has your results?

A previous draft had results and discussion separated and the feedback from several readers was that this caused unnecessary confusion, duplication and jumping back and forth. We will keep the results and discussion merged.

270: I'm confused here, sorry. Weren't the ALR measurements entirely in the warm inflow, since they were so deep?

We will clarify this sentence, you are correct that all our ALR measurements are in the warm layer of in the cavity, but we define the inflow as the narrow bottom intensified current along the 700 m isobath.

272: runon sentence.

The sentence in line 272 is not that long, are you sure that this is the line number you meant?

273: Due to what mechanism? We will add this information in the revise manuscript.

281: Please change "this" to "their" to avoid confusing with your study.

we will make this change in the revised manuscript.

285: If you are going to state dissipation rates this low, I think you do need to demonstrate your minimum detectability threshold. Earlier you said it was 1e-10. So how then do you get a median lower than this.

The detection limit is between $10^{-11}$ and $10^{-10}$ depending on the dive track. We never state that the detection limit is $10^{-10}$ and will clarify the sentence you refer to.

Again, I think median should be avoided for all quantities unless there is a good reason. Why not just use the mean?

We do not want to cause confusion by switching between mean and median for data with and without outliers or non-normal distributions, since median(x) = mean(x) when x is normally distributed we think median is a better choice.

333: The reason for these calculations is revealed here - suggest giving it earlier to make the reader understand why they are being told all of this. More fundamentally, is that the only reason turbulence is important to measure under ice shelves? Ie, as a possible mitigator of the advective heat flux by these warm flows?

We will move the motivation for the mixing calculation to earlier in the paper. As far as the ice melt rate and modelling efforts in the cavity are concerned the heat flux is the major concern. As we discuss above, mixing is also important for the trace metal and nutrient transport, however we do not have measurements of concentration gradients in the cavity and can not make a statement as to how the mixing influences them.

I, at least as a non ice sheet person, would like to see a cartoon (words or actual graphic) showing a cross section of the hypothesized warm water flow to the grounding line. The reason for this is that I don't currently understand why the study focused so much on the near-bottom mixing. I'd think that the heat loss out of the mCDW would be better quantified near its upper edge. As the authors point out, the water near the bottom is very weakly stratified so the heat fluxes are expected to be small. Aloft nearer the interface, the gradients would be stronger, but also the distance from the topography which is presumably generating most of the turbulence (my comments above about that not having been adequately demonstrated notwithstanding). So, statements that mixing is weak such as on lines 356-258 should be tempered somewhat. And I think the cartoon or written description of the flow giving readers the sense of which depths are thought the most likely to eventually contact the ice would help inform this discussion, at least for me.

We will add a cartoon to the revised manuscript, thank you for the suggestions. As a short description of why the bottom layer of mCDW is so important is that the grounding line (the region where the ice flow contacts the ocean and comes afloat) is the area where most melting and glacier retreat takes place.

**References**

Davis, P. E. D., Jenkins, A., Nicholls, K. W., Dutrieux, P., Schröder, M., Janout, M. A., . . . McPhail, S. (2022, November). Observations of Modified Warm Deep Water Beneath Ronne Ice Shelf, Antarctica, From an Autonomous Underwater Vehicle. *Journal of Geophysical Research: Oceans*, *127*(11). doi: 10.1029/2022jc019103

Polzin, K., Kunze, E., Hummon, J., & Firing, E. (2002). The finescale response of lowered ADCP velocity profiles. *Journal of Atmospheric and Oceanic Technology*, *19*(2), 205–224. doi: 10.1175/1520-0426(2002) 019⟨0205:tfrola⟩2.0.co;2

Wåhlin, A., Alley, K. E., Begeman, C., Hegrenæs, Ø., Yuan, X., Graham, A. G. C., . . . Heywood, K. J. (2024, August). Swirls and scoops: Ice base melt revealed by multibeam imagery of an Antarctic ice shelf. *Science Advances*, *10*(31). doi: 10.1126/sciadv.adn9188

---

## Author Comment (AC2)

**Response to reviewer R1 comments**

August 22, 2025

Thank you for your detailed and helpful review. In this document, reviewer comments are in **black** and our comments are in **red**. New text added to the manuscript is in blue.

This is a clearly written paper with nice figures describing nice analysis of an extraordinarily rare and hard to obtain dataset. The manuscript should be published.

Thank you for your positive assessment of our paper, we will address your individual points below

I do have a number of comments, questions and morsels for thought that I list below in the order in which I read. The majority are (very) minor, amounting to text and grammar nits, but some are more substantive. In particular I would like to see

- more supporting evidence behind the claim that mixing is weak (for the reasons given in the final comment below),

  We show that the median TKE dissipation rate is $10^{-11}$ to $10^{-10}$, which are very low values, comparable to the background TKE dissipation rate in the ocean (see Figures 6 and 7 in Waterhouse et al. (2014)) Waterhouse et al. (2014) does not give average values for epsilon, instead we can compare values for kappa. In our study median values of kappa

range between $0.2 \times 10^{-4} m^2 s^{-2}$ – $1.1 \times 10^{-4} m^2 s^{-2}$. Waterhouse et al. (2014) gives average deep ocean (depth between 1000 m and the bottom) values of kappa as $4.3(0.4 - 11.5) \times 10^{-4} m^2 s^{-1}$, with the values in parenthesis the 95-th percentile bootstrap confidence range. This indicates that our values of kappa lie within the lower range or just below the global distribution for the deep ocean. We will add a reference to global average values of mixing in Waterhouse et al. (2014). Additionally, we will include a new figure showing the distribution of measured epsilon in different ice shelf cavities. Our values lie within the range of previous observations as we have thus rephrased our abstract to remove the reference to "low mixing", the sentence in question now reads "Rates of background mixing are $\varepsilon \approx 10^{-10} \ W \ kg^{-1}$ with patches of higher mixing of $\varepsilon \approx 10^{-8} \ W \ kg^{-1}$."

- better figure 3 and 4, which currently mixes aspect ratios, has the reader going back and forth and does not allow direct comparisons of the most relevant quantities - specifically epsilon and the different instability indicators

We have combined Figures 3 and 4 to the new Figure 3. All panels now have the same aspect ratio.

- quantification of the ADCP vertical wavenumber response and hence justification of the numerical values of Ri presented (or alternately toning down the reference to specific values such as $Ri = 1/4$ given the estimates are noisy and not fully resolved),

We are unsure what the reviewer is asking about here. Do you refer to the vertical wave number response that Polzin et al. (2002) refer to when estimating turbulent mixing processes from vertical shear in the ADCP? We do not use the ADCP to calculate mixing, we only use it to get information on the horizontal velocity in the vicinity of our microstructure shear measurements. The Richardson number is calculated from the vertical shear between successive 8 m tall (in the vertical) ADCP bins, but this is not used for the turbulent shear calculations.

- justification for use of median versus mean

  We use the median as it is less impacted by outliers or non-normal distribution of values. If the data is normally distributed the median and mean are identical, so there is no negative effect of using the median as the default method for averaging values.

- and finally and perhaps most substantively, an explanation for why the turbulent heat fluxes just above the bottom are important to measure. Ie, is that the water that will eventually meet the grounding line, or should the study have been done nearer the top of the mCDW watermass where the gradients and heat losses are much stronger?

  Have added information to the text to clarify that we measure heat fluxes close to the bottom to capture the effect of topography roughness on the flow, to capture the mixing where the bottom intensified warm inflow interacts with the seabed and due to practical constraints (the ALR needs to stay within 100 m of the seabed to allow for accurate dead-reckoning and bottom tracking). The schematic we added to the Introduction (see your comment below) should also make the reason for our interest in the lower mCDW clearer. We have added the following sentence to the introdution: "our study targets the current of warm mCDW flowing into the ice shelf cavity and maintains a dive track close to the seabed. We investigate the circulation and mixing in the mCDW inflow close to the bed of the cavity to understand the effect of bathymetry on mixing and circulation. We quantify the upward heat transport that cools the mCDW in the deepest part of the cavity whilst warming the overlying mCDW (which can access the grounding line and the ice shelf base; Figure 1), and investigate drivers for the observed mixing. "

Good luck. I enjoyed reading the paper and hope that these comments are useful.

11: topography, turbulent or both not resolved?

We have clarified this sentence to confer that turbulent mixing is not resolved in models and topography is not resolved in bathymetry products or models. The sentence now reads: "We show a highly complex spatial pattern of turbulent mixing and of bottom topography. The bottom topography is currently not resolved in bathymetry products and both the topography and turbulent mixing are currently not resolved in models of ice-shelf–ocean interactions."

26: awkward

The sentence now reads " The mCDW can cause melting at the grounding line, leading to basal mass loss and grounding line retreat."

35, 53: "this" is a weak reference. Please reword; see Strunk and White if needed.

the sentences now read " The depth at which meltwater enters the ocean is influenced by where melt predominantly occurs. "and "Due to the remote location and difficult access, measuring turbulent kinetic energy dissipation rate in ice shelf cavities is only now starting to become feasible.", respectively.

48: Melt rates two words?

We have corrected this

52: This statement is actually not true: epsilon is the dissipation rate and further assumptions must be invoked to infer the mixing. This needs to be corrected and expanded upon.

We have clarified that $\varepsilon$ is only a measure of turbulence if the turbulence is isotropic. The sentence now reads "The turbulent kinetic energy dissipation rate, $\varepsilon$, is the rate at which molecular viscosity dampens isotropic turbulence generated at large scales by e.g. vertical or lateral shear, and is used to quantify turbulent mixing."

55: This would be a good place to distinguish what is different about this study from the other two.

Thank you for your comment, the paragraph in question now reads: "To our knowledge, there exist two published studies of mixing in an ice-shelf cavity measured by an underwater vehicle, one under Pine Island Glacier (Kimura et al., 2016), and one under the Filchner Ronne Ice Shelf (Davis et al., 2022). We present a third such study, targeting DIS. DIS and Pine Island Ice Shelf experience low tidal flows, whereas Filchner Ronne Ice Shelf experiences strong tidal flows. Unlike Davis et al. (2022) and Kimura et al. (2016), our study targets the current of warm mCDW flowing into the ice shelf cavity and maintains a dive track close to the seabed. We investigate the circulation and mixing in the mCDW inflow close to the bed of the cavity to understand the effect of bathymetry on mixing and circulation. We quantify the upward heat transport that cools the mCDW in the deepest part of the cavity whilst warming the overlying mCDW (which can access the grounding line and the ice shelf base; Figure 1), and investigate drivers for the observed mixing. "

56: which − > that. Also, is this the only reason mixing is important to know for these situations?

Thank you, we have corrected that. Mixing at the seabed – ocean interface is also important for nutrient transport, such as the transport of iron from sedimentary sources to the euphotic zone. We refer to such processes in the paragraph above: "The input of meltwater to the Amundsen Sea is also important for biological activity in the region. The flow of mCDW along the seafloor on its way into the DIS cavity enriches the mCDW in dissolved iron and manganese while the meltwater from the ice shelf itself is a source of particulate iron and manganese (van Manen et al., 2022). The addition of glacial meltwater makes the outflowing mCDW more buoyant than the dense mCDW infow, transporting iron and manganese to the surface ocean (van Manen et al., 2022) where they are important micronutrients for primary producers (Twining & Baines, 2013)."

66-68: Please give order of magnitude of the clock offsets before correction and the precision of the alignment afterwards.

We have added this information in the revised manuscript. The paragraph now reads: "A clock offset of approximately 2 minutes between the ALR CTD and the MicroRider was resolved by calculating lagged correlations between the MicroRider pressure sensor and the CTD pressure sensor to find the offset, then correcting for the identified clock offset and drift. ".

74: Please explain why you used median instead of mean?

See our explanation above.

95: Could indicate this is likely because of $F = ma$; ie the same force on the huge autos produces much smaller accelerations.

Thank you for this prompt, the revised sentence now reads: " Unlike microstructure measurements performed with a small, light-weight AUV (e.g. Kolås et al., 2022), the shear microstructure recorded on AutoSub Long Range was not critically impacted by vehicle vibrations, possibly due to its greater mass."

105: on which this study focuses.

Thank you, we have made the correction.

105 general: is this the first paper that presents the details of shear microstructure from Autosub? Surprising if so but if true, you might consider showing a few spectra and additional details, possibly in an appendix, so that future work can cite this paper.

This is not the first such paper, we refer the reader to Davis et al. (2022) for information of the spectral response of the shear probes on ALR. We have added the sentence "The shear power spectra from a MicroRider mounted on an ALR have been described in detail in Davis et al. (2022)." to the manuscript.

111: Shih et al is a very bad reference for this! They find a $Re_b$-dependent Gamma. Suggest just citing Osborn (1980). There are also now a handful of observational references supporting the assertion that $\Gamma = 0.2$.

Thank you for pointing this out, we have removed the reference to Shih. The sentence now reads "$\Gamma = 0.2$ is the mixing efficiency, a measure of the amount of available turbulent kinetic energy that is permanently converted to potential energy by turbulent mixing, which is generally set to 0 2 (Osborn, 1980)"

113: How close to the bottom of the ice is the shallowest CTD measurement shown? The very strong gradients at the very top of the cavity CTD casts (Fig 2 black) are interesting.

The CTD cast goes right to the ice – ocean interface. We refer the reader to A. Wåhlin et al. (2024) for a discussion of the CTD measurements at the interface.

123 and throughout: I believe units should be in roman, not italicized, font.

We have corrected this where we found such instances, all remaining formatting will be finalized in the copy editing process.

136: Suggest reformatting the equation.

We have reformatted the equation.

140: Please make it very clear that Ri (under the ice at least) is based on a single N2 profile whereas the shear is a function of location and time. This is OK, but appropriate caveats as to its governing local instabilities without in-situ N2 should be given.

Thank you for this comment, we have added the following words to the paragraph describing Ri: "Thus, Ri is calculated from a constant value of $N^2$, based on a single profile in the cavity, and shear is a function of space and time along the track of the ALR. Variations of Ri due to variations in $N^2$ are not captured. For constant $N^2$, Ri is low in areas of high shear."

173: Generally, avoid "there is" in favor of more active language such as "flow is to the . . . "

We will change some of our wording where we deem appropriate in the revised manuscript. The sentence in question here has been reworded to "A bottom intensified southward current flows into the cavity in the east, between the 400 m and 900 m isobaths, and a shallower, bottom intensified northward current flows out of the cavity in the west (Figure 3c)."

177: High compared to what?

This sentence has been rephrased to read "Below 500 m depth, turbulent kinetic energy dissipation is elevated (compared to other areas below 500 m along the ice front) in the inflow. Turbulent kinetic energy dissipation is $\approx 10^{-8}\,\mathrm{W\,kg^{-1}}$ in the inflow over an area approximately 7 km wide and 200 m high (Figure 3d; turbulent kinetic energy dissipation rate is elevated between 38 km and 45 km of the ice front and $\sim 200\,\mathrm{m}$ above the seabed)."

177: runon sentence.

In addition to adding context (see above), this sentence has been split into shorter sentences.

Figure 3, lines 2 and 4 of caption: runon sentences. Also, the dots are said to indicate the starting locations - but they are a continuous line. I'd have thought there would just be two starting locations, one for center and one for east? Please clarify.

The new Figure 3 has shorter sentences in the caption and the dots are described as: "10-minute medians of the values measured by the ALR are shown as coloured dots in panels a-d. The two dots with bold outlines show the starting locations of the ALR east and centre short dive tracks into the cavity."

Figure 4: Personally I think it would be better to keep the aspect ratio constant between Fig 3 and 4. Also, sine you already plotted velocity in Figure 3, suggest including a panel of N2. The aspect ratio is all the more a problem later when the authors are comparing epsilon to the different instability indicators - but the reader must go back and forth between figure

3 and 4. Suggest standardizing the aspect ratio and including an epsilon panel in Figure 4. Possibly even adding Ri contours to the epsilon panel or epsilon contours to the Ri panel since the authors are trying to demonstrate correspondence between the two quantities.

Thank you for this feedback, we have combined Figure 3 and 4 into the new Figure 3, in which all panels have the same aspect ratio. We have also included a panel of N2 at the ice front. We have not plotted Ri contours on the epsilon panel, as that proved to be confusing (switching between density contours and Ri contours).

Also, the Ri panel is just a big sea of red. Consider plotting something else to highlight the unstable regions such as $Ri^{-1}$ or $Fr = Uz/N$.

The Ri panel is mainly red due to the choice of colourbar. We chose to plot $Ri < 1/4$, $1 > Ri > 1/4$, and $Ri > 1$ as three different colours in keeping with established practice (e.g. Dotto et al., 2025) to distinguish along criteria for instability. Plotting 1/Ri would make it less obvious where $Ri < 1/4$. We would like to avoid plotting additional instability metrics such as the Froude number to avoid confusion.

182: Doesn't negative PV mean unstable? The whole water column is unstable? Is it backwards in the southern hemisphere? Some statements to clarify would be useful.

We have clarified this in the text by adding the sentence: "Instabilities may develop when potential vorticity and $f$ have opposite signs, as $f$ is negative in the southern hemisphere, potential vorticity $> 0$ indicates conditions favourable to instability. "

188: I don't agree with this statement - the high dissipation does not appear to me to line up at all with for Ri. Furthermore, given the ADCP's finite vertical resolution and noise, some additional detail needs to be given on how seriously we are to take the numerical value of Ri. I think that either some wavenumber spectra and transfer functions a la Polzin 2002 need to be

included, or Ri used as a qualitative indicator.

As far as we understand Polzin et al., 2002 the vertical wavenumber response of the ADCP is relevant when calculating turbulent dissipation from the ADCP. We are not using the ADCP for turbulence. We use the VMP or microrider for shear microstructure and the LADCP and ADCP on the ALR to get an idea of the vertical and horizontal structure of the water column at much larger scales, a background value if you will. Ri and other stability criteria are frequently calculated from LADCP output with bin sizes of 8 m and used in comparisons with microstructure data (e.g. Dotto et al., 2025; Naveira Garabato et al., 2017; Naveira Garabato et al., 2019). We have clarified our reference to Ri and mixing, the paragraph now reads:"The region of high turbulent kinetic energy dissipation rate $\varepsilon$ in the inflow (Figure 3d) coincides with instances of $R_i < 1/4$ captured at $40\,\text{km}$ (Figure 3h), indicating conditions favourable to turbulent mixing. Turbulent kinetic energy dissipation rate is larger than $10^{-8}$ here, one to two orders of magnitude higher than the background value (Figure 3d). Dotto et al. (2025) found similar results for the outflow of DIS. Although areas of high $\varepsilon$ extend beyond areas of Ri $< 1/4$, $\varepsilon$ is higher and Ri is lower in the upper watercolumn and close to the seabed. We observe areas of low Ri and Ri $< 1/4$ that are not associated with high values of $\varepsilon$, e.g. at $25\,\text{km}$ along the transect. "

191: I disagree; elevated mixing is much broader than the regions of $Ri < 1/4$ - augmenting my previous point.

We will clarify that the high epsilon includes, but extends beyond, the region of low Ri. The relevant sentence now reads:"Although areas of high $\varepsilon$ extend beyond areas of Ri $< 1/4$, $\varepsilon$ is higher and Ri is lower in the upper watercolumn and close to the seabed."

193: This statement is not justified. Epsilon appears surface intensified as well. And while it is bottom intensified, I do not think the statement that it is heightened over rough topography, shear or high currents (of which you generally must choose either high current or high shear, not both...) is supported. And as before, I don't think that high epsilon lines up with low

Ri either. Either way, if this statement is retained, more analysis needs to be shown - scatter plots, binned averages, etc.

We have clarified that we are only considering epsilon below the Winter Water layer, thus we do not discuss high epsilon at the surface. We have included the following: "Below 500 m depth, turbulent kinetic energy dissipation is elevated in the inflow (compared to other areas below 500 m along the ice front). "With regards to the ice shelf front, we have changed our statement to read: "Our observations show turbulent mixing to be patchy, bottom intensified and to coincide with high velocities (Figure 3)."We maintain that in the cavity high epsilon is associated with high shear and low Ri and have added correlations plots that show this to the manuscript. See more below.

197: runon sentence. And seemingly unrelated sentences. Ri governs shear instability, not symmetric instability... (I understand they are highly correlated here, but they are different, so clarification is needed).

We will insert a paragraph break before "at the nearby Pine Island Ice Shelf....". The start of the new paragraph now reads: " At the nearby Pine Island Ice Shelf (PIIS) Naveira Garabato et al. (2017) conducted ADCP and VMP transects along the calving front. Naveira Garabato et al. (2017) do not detect a fast, narrow, turbulent inflow current, unlike what we observed at DIS (Figure 3). High rates of turbulent kinetic energy dissipation below the WW were mostly confined to the PIIS outflow. The PIIS is connected to another ice shelf cavity to the north and may receive some of its inflow from under this neighbouring ice shelf, which may decrease the inflow across the PIIS front and possibly the turbulent mixing there. Additionally, the ice shelf draft of the PIIS is deeper ($\approx 400$ m) than the DIS ($\approx 350$ m). The ice shelf draft induces a barotropic jump (an abrupt change in water column thickness, blocking flow along constant lines of water column thickness) and limits barotropic inflow to the cavity (A. K. Wåhlin et al., 2020), thus decreasing inflow current velocities and possibly turbulent mixing. ". The sentence regarding Ri has been removed, the relevant paragraph now reads "Symmetric instability is driven by high vertical current shear (Figure 3j). The region of

high turbulent kinetic energy dissipation rate $\varepsilon$ in the inflow (Figure 3d) coincides with instances of $R_i < 1/4$ captured at $40\,\mathrm{km}$ (Figure 3h), indicating conditions favourable to turbulent mixing."

202: What is a barotropical jump?

It is an oceanographic term for an abrupt change in water column thickness. This occurs at the ice shelf front, since ocean currents want to flow along lines of uniform water column thickness, the ice shelf draft poses a barrier to flow, even at depths deeper than its draft. We have have added a parenthetical "The ice shelf draft induces a barotropic jump (an abrupt change in water column thickness, blocking flow along constant lines of water column thickness) and limits barotropic inflow to the cavity (A. K. Wåhlin et al., 2020), thus decreasing inflow current velocities and possibly turbulent mixing. "to the sentence in question.

207: Please rewrite this passive and vague sentence.

We have rephrased this sentence, it now reads "Because the ALR measurements were not coincident in time with the LADCP section, the ALR may have failed to capture transient patches of high turbulent kinetic energy dissipation rate present in the LADCP section."

204-210: Suggest moving this speculative bit to the discussion.

We originally had results and discussion split, but chose to integrate them to avoid duplicating information and to limit jumping back and forth between topics. We will retain this structure.

216: I think it would be nice to compare this to open ocean values at a similar depth and/or abyssal values, for context. Otherwise "weakly stable" doesn't have meaning.

We have added typical open ocean values for N2 in the Southern Ocean. The sentence now reads: "We estimate $N^2$ below a depth of $900\,\mathrm{m}$ to be $6 \times 10^{-7}\mathrm{s}^{-1}$. This is about three orders of magnitude lower than typical open

ocean values for the southern ocean (King et al., 2012), indicating weakly stable stratification in the cavity. "

218: Style guides such as Strunk and White suggest avoiding "Figure x shows..." in favor of "statement x is true (Figure y)."

We have changed this sentence to read: "In the cavity, the ALR detected currents that flow predominantly southeastward with low vertical shear in the east dive track, and a more mixed pattern in the two centre dive tracks (Figures 5 and 6)."

223: Figure 6 and 5 -> Figures 5 and 6

Thank you for pointing out this typo, it has been corrected.

236: redundant. Suggest "Maximum values were" or "Values reached."

We have changed the sentence to read: "Maximum values were $10^{-7}\,\mathrm{W\,kg^{-1}}$ ($\varepsilon$) and $10^{-2}\,\mathrm{m^2\,s^{-1}}$ ($\kappa$), respectively (Table 2)."

238: Again, I'm afraid I don't see this. There are counter examples where epsilon is high over flat bottoms. Please include plots that allow direct comparison such as plotting epsilon with Ri, current speed or bathymetric slope over plotted, or scatter plots or binned averages (e.g. epsilon(Ri) etc) if you want to make this claim.

We have included scatter plots in our revised manuscript. We stress that our data are extremely noisy and thus correlation coefficients are low even if relationships are statistically significant to the 0.1% level. Additionally, the bathymetry gradient deeper in the cavity is affected by the low resolution of BedMachine, preventing us from fully resolving the relationship between bathymetry and epsilon.

241: Please remind reader that it's Ri computed from in-situ where and

The remainder of this sentence seems to be missing, please clarify

245: Again, please include transfer function and instrument response information if you wish to quantify the numerical value of Ri versus using it as a qualitative indication. Note as well that these transfer functions and hence the mapping of true to measured Ri will be different for the Autosub and the LADCP.

Can you clarify what you mean by the mapping of true to measured Ri and how that is influenced by the ADCP? It is common practice to calculate Ri from vertical shear from the 8 m binned ADCP and we do not use the ADCP to calculate fine scale microstructure.

257: Is it really necessary to use a package like this to compute a spatial gradient? More fundamentally I do not see a relationship between RMS bathymetric slope and dissipation rate.

We do not use a package to calculate the gradient. The bathymetry from ALR is only 1D, to get a 2D gradient we use BedMachine to get the bathymetry normal to and along the ALR dive track. Can you clarify what you mean by "RMS bathymetric slope"? We do not calculate RMS of the slope and to our eyes there is a clear relationship between the bathymetry and epsilon close to 0 km on the east dive track. We have included scatter plots, linear fit lines, correlation coefficients and p-values for the relationship between bathymetric slope and epsilon which clearly show a strong connection.

264 onwards: consider moving all of this comparison to past work to the discussion, so that the results section just has your results?

A previous draft had results and discussion separated and the feedback from several readers was that this caused unnecessary confusion, duplication and jumping back and forth. We will keep the results and discussion merged.

270: I'm confused here, sorry. Weren't the ALR measurements entirely in the warm inflow, since they were so deep?

We will clarify this sentence, you are correct that all our ALR measurements are in the warm layer of in the cavity, but we define the inflow as the narrow bottom intensified current along the 700 m isobath. The sentence now reads:

"We observed our highest mixing values in the bottom intensified inflow to the cavity, whereas Kimura et al. (2016) observed the highest levels of mixing close to the grounding line. Our ALR dive tracks did not reach the grounding line, and the dive tracks of Kimura et al. (2016) did not cover the inflow of the PIIS, making comparison difficult. Naveira Garabato et al. (2017) did not find enhanced mixing in the PIIS inflow. "

272: runon sentence.

The sentence in line 272 is not that long, are you sure that this is the line number you meant?

273: Due to what mechanism? This sentence has been modified to read: "Kimura et al. (2016) hypothesised that high (horizontal) density gradients driven by temperature differences and a bathymetric ridge can drive a baroclinic current with strong vertical current shear. This high shear in turn drives high levels of turbulence at the ridge under PIIS. Our study shows that high density gradients are not a requirement for high levels of turbulence."

281: Please change "this" to "their" to avoid confusing with your study.

This change has been made.

285: If you are going to state dissipation rates this low, I think you do need to demonstrate your minimum detectability threshold. Earlier you said it was 1e-10. So how then do you get a median lower than this.

The detection limit is between $10^{-11}$ and $10^{-10}$ depending on the dive track. We never state that the detection limit is $10^{-10}$ and have clarified the sentence you refer to. The paragraph now reads: "Smaller, narrower peaks at frequencies below 10Hz in the accelerometer spectra are successfully removed by the Goodman method for dissipation rates above $1 \times 10^{-8}\,\mathrm{W\,kg^{-1}}$. Deviations from the fitted Nasmyth spectra remain for dissipation rates below $1 \times 10^{-9}$, arguing that quantitative estimates of dissipation rate in very quiescent regimes are not as reliable as estimates of high dissipation rates. Individual dive tracks show good agreement between shear spectra and Nasmyth spectra for dissipation rates lower than $1 \times 10^{-10} \, \mathrm{W \, kg^{-1}}$. Where dissipation rates calculated from two orthogonal shear probes show good agreement, we are confident in reporting dissipation rates down to $1 \times 10^{-11} \, \mathrm{W \, kg^{-1}}$. Additionally, any signal in the shear spectra caused by the AUV motion, and not removed by the Goodman filter, will have minimal effects on the spatio-temporal pattern of high and low $\varepsilon$ observed by the ALR or the qualitative assessment of these patterns, on which this study focuses."

Again, I think median should be avoided for all quantities unless there is a good reason. Why not just use the mean?

We do not want to cause confusion by switching between mean and median for data with and without outliers or non-normal distributions, since median(x) = mean(x) when x is normally distributed we think median is a better choice.

333: The reason for these calculations is revealed here - suggest giving it earlier to make the reader understand why they are being told all of this. More fundamentally, is that the only reason turbulence is important to measure under ice shelves? Ie, as a possible mitigator of the advective heat flux by these warm flows?

We have added the following paragraph to the introduction together with a scematic of the ice shelf cavity: "Basal melt under Dotson is highest close to the grounding line of the Kohler East (often referred to as Smith West) and Kohler West glaciers (Khazendar et al., 2016; Gourmelen et al., 2017). The Kohler West grounding line lies at the southern end of the dashed path shown in Figure 1a. A cross-section of the cavity along the path (Figure 1b) shows an idealized view of the cavity circulation under the Dotson Ice Shelf. Warm water entering the cavity in the east, and traveling along a path shallower than the 830 m deep sill (Jordan et al., 2020), can reach the grounding line. Warm water that reaches the grounding line causes high basal melt and grounding line retreat (Khazendar et al., 2016; Gourmelen et al., 2017). The sill may limit direct access of the deepest and warmest mCDW to the

grounding line (Jordan et al., 2020; Khazendar et al., 2016). The addition of meltwater to the warm, salty mCDW forms a buoyant plume which travels along the underside of the ice before exiting the cavity in the west. Along its path, the water experiences turbulent mixing which can transport heat and salt upward, modifying the properties of the water which ultimately interacts with the grounding line, and the properties of the buoyant plume exiting the cavity." As far as the ice melt rate and modelling efforts in the cavity are concerned the heat flux is the major concern. As we discuss above, mixing is also important for the trace metal and nutrient transport, however we do not have measurements of concentration gradients in the cavity and can not make a statement as to how the mixing influences them.

I, at least as a non ice sheet person, would like to see a cartoon (words or actual graphic) showing a cross section of the hypothesized warm water flow to the grounding line. The reason for this is that I don't currently understand why the study focused so much on the near-bottom mixing. I'd think that the heat loss out of the mCDW would be better quantified near its upper edge. As the authors point out, the water near the bottom is very weakly stratified so the heat fluxes are expected to be small. Aloft nearer the interface, the gradients would be stronger, but also the distance from the topography which is presumably generating most of the turbulence (my comments above about that not having been adequately demonstrated notwithstanding). So, statements that mixing is weak such as on lines 356-258 should be tempered somewhat. And I think the cartoon or written description of the flow giving readers the sense of which depths are thought the most likely to eventually contact the ice would help inform this discussion, at least for me.

Have added a schematic to the revised manuscript, thank you for the suggestion. We have added the following text to the introduction: "Basal melt under Dotson is highest close to the grounding line of the Kohler East (often referred to as Smith West) and Kohler West glaciers (Khazendar et al., 2016; Gourmelen et al., 2017). The Kohler West grounding line lies at the southern end of the dashed path shown in Figure 1a. A cross-section of the cavity

along the path (Figure 1b) shows an idealized view of the cavity circulation under the Dotson Ice Shelf. Warm water entering the cavity in the east, and traveling along a path shallower than the 830 m deep sill (Jordan et al., 2020), can reach the grounding line. Warm water that reaches the grounding line causes high basal melt and grounding line retreat (Khazendar et al., 2016; Gourmelen et al., 2017). The sill may limit direct access of the deepest and warmest mCDW to the grounding line (Jordan et al., 2020; Khazendar et al., 2016). The addition of meltwater to the warm, salty mCDW forms a buoyant plume which travels along the underside of the ice before exiting the cavity in the west. Along its path, the water experiences turbulent mixing which can transport heat and salt upward, modifying the properties of the water which ultimately interacts with the grounding line, and the properties of the buoyant plume exiting the cavity."

**References**

Davis, P. E. D., Jenkins, A., Nicholls, K. W., Dutrieux, P., Schröder, M., Janout, M. A., . . . McPhail, S. (2022, November). Observations of Modified Warm Deep Water Beneath Ronne Ice Shelf, Antarctica, From an Autonomous Underwater Vehicle. *Journal of Geophysical Research: Oceans*, *127*(11). doi: 10.1029/2022jc019103

Dotto, T. S., Sheehan, P. M. F., Zheng, Y., Hall, R. A., Damerell, G. M., & Heywood, K. J. (2025, May). Heterogeneous Mixing Processes Observed in the Dotson Ice Shelf Outflow, Antarctica. *Journal of Geophysical Research: Oceans*, *130*(5). doi: 10.1029/2024jc022051

Gourmelen, N., Goldberg, D. N., Snow, K., Henley, S. F., Bingham, R. G., Kimura, S., . . . van de Berg, W. J. (2017, October). Channelized Melting Drives Thinning Under a Rapidly Melting Antarctic Ice Shelf. *Geophysical Research Letters*, *44*(19), 9796–9804. doi: 10.1002/2017gl074929

Jordan, T. A., Porter, D., Tinto, K., Millan, R., Muto, A., Hogan, K., . . . Paden, J. D. (2020, September). New gravity-derived bathymetry for the

Thwaites, Crosson, and Dotson ice shelves revealing two ice shelf popula-
tions. *The Cryosphere*, *14*(9), 2869–2882. doi: 10.5194/tc-14-2869-2020

Khazendar, A., Rignot, E., Schroeder, D. M., Seroussi, H., Schodlok, M. P.,
Scheuchl, B., ... Velicogna, I. (2016, October). Rapid submarine ice
melting in the grounding zones of ice shelves in West Antarctica. *Nature
Communications*, *7*(1). doi: 10.1038/ncomms13243

Kimura, S., Jenkins, A., Dutrieux, P., Forryan, A., Naveira Garabato,
A. C., & Firing, Y. (2016, December). Ocean mixing beneath Pine Is-
land Glacier ice shelf, West Antarctica: OCEAN MIXING BENEATH
PIG. *Journal of Geophysical Research: Oceans*, *121*(12), 8496–8510. doi:
10.1002/2016jc012149

King, B., Stone, M., Zhang, H. P., Gerkema, T., Marder, M., Scott, R. B.,
& Swinney, H. L. (2012, April). Buoyancy frequency profiles and internal
semidiurnal tide turning depths in the oceans. *Journal of Geophysical
Research: Oceans*, *117*(C4). doi: 10.1029/2011jc007681

Kolås, E. H., Mo-Bjørkelund, T., & Fer, I. (2022, March). Technical note:
Turbulence measurements from a light autonomous underwater vehicle.
*Ocean Science*, *18*(2), 389–400. doi: 10.5194/os-18-389-2022

Naveira Garabato, A. C., Forryan, A., Dutrieux, P., Brannigan, L., Biddle,
L. C., Heywood, K. J., ... Kimura, S. (2017, January). Vigorous lat-
eral export of the meltwater outflow from beneath an Antarctic ice shelf.
*Nature*, *542*(7640), 219–222. doi: 10.1038/nature20825

Naveira Garabato, A. C., Frajka-Williams, E. E., Spingys, C. P., Legg, S.,
Polzin, K. L., Forryan, A., ... Meredith, M. P. (2019, June). Rapid
mixing and exchange of deep-ocean waters in an abyssal boundary current.
*Proceedings of the National Academy of Sciences*, *116*(27), 13233–13238.
doi: 10.1073/pnas.1904087116

Osborn, T. R. (1980, January). Estimates of the Local Rate of Vertical Dif-
fusion from Dissipation Measurements. *Journal of Physical Oceanography*,
*10*(1), 83–89. doi: 10.1175/1520-0485(1980)010⟨0083:eotlro⟩2.0.co;2

Polzin, K., Kunze, E., Hummon, J., & Firing, E. (2002). The finescale response of lowered ADCP velocity profiles. *Journal of Atmospheric and Oceanic Technology*, *19*(2), 205–224. doi: 10.1175/1520-0426(2002) 019⟨0205:tfrola⟩2.0.co;2

Twining, B. S., & Baines, S. B. (2013, January). The Trace Metal Composition of Marine Phytoplankton. *Annual Review of Marine Science*, *5*(1), 191–215. doi: 10.1146/annurev-marine-121211-172322

van Manen, M., Aoki, S., Brussaard, C. P., Conway, T. M., Eich, C., Gerringa, L. J., ... Middag, R. (2022, October). The role of the Dotson Ice Shelf and Circumpolar Deep Water as driver and source of dissolved and particulate iron and manganese in the Amundsen Sea polynya, Southern Ocean. *Marine Chemistry*, *246*, 104161. doi: 10.1016/j.marchem.2022 .104161

Wåhlin, A., Alley, K. E., Begeman, C., Hegrenæs, Ø., Yuan, X., Graham, A. G. C., ... Heywood, K. J. (2024, August). Swirls and scoops: Ice base melt revealed by multibeam imagery of an Antarctic ice shelf. *Science Advances*, *10*(31). doi: 10.1126/sciadv.adn9188

Wåhlin, A. K., Steiger, N., Darelius, E., Assmann, K. M., Glessmer, M. S., Ha, H. K., ... Viboud, S. (2020, February). Ice front blocking of ocean heat transport to an Antarctic ice shelf. *Nature*, *578*(7796), 568–571. doi: 10.1038/s41586-020-2014-5

Waterhouse, A. F., MacKinnon, J. A., Nash, J. D., Alford, M. H., Kunze, E., Simmons, H. L., ... Lee, C. M. (2014, July). Global Patterns of Diapycnal Mixing from Measurements of the Turbulent Dissipation Rate. *Journal of Physical Oceanography*, *44*(7), 1854–1872. doi: 10.1175/jpo-d-13-0104.1

---

## Author Comment (AC3)

**Response to reviewer R2 comments**

August 22, 2025

Thank you for your detailed and helpful review. In this document, reviewer comments are in **black** and our comments are in **red**. New text added to the manuscript is in blue.

This paper discusses microstructure observations made beneath Dotson Ice Shelf using an Autonomous Submersible Vehicle. The data appear to lack the temporal and spatial coverage that would enable substantive conclusions to be drawn about the role of turbulent mixing in the larger-scale processes that operate beneath the ice shelf. They are, nevertheless, intrinsically interesting, in that they represent some of the very few direct observations that we have from within a sub-ice-shelf cavity. That remote part of the ocean plays a pivotal role in setting the mass balance of the Antarctic Ice Sheet and hence its impact on global sea level, so any observations are of value. I would therefore recommend publication of the paper with only relatively minor changes.

Thank you for your positive assessment of our manuscript. We agree that a greater spatial and temporal range of observations in ice shelf cavities is needed to gain a complete picture of the water mass transformations, heat and (fresh)water transport that influence Antarctic ice mass loss, grounding line retreat, sea level rise, deep water formation and nutrient transport. Until such measurements are routinely possible, we intend our manuscript to offer a glimpse at conditions and possible processes.

Suggested changes:

Title:

It's a minor point, but the current title does not reflect the content of the paper very well. It promises observations of ocean currents. While they are included there is very little discussion of them, and no more space is devoted to currents than to water properties.

We have changed the title of the paper to: "Observations of turbulent mixing in the Dotson Ice Shelf cavity"

Abstract:

Reflects the content of the paper and thus its main weakness, which is a lack of substantive conclusions. I accept that it is hard to put such detailed observations into a broader context, especially when they are made in such a data-poor region. However, I wonder if it might be possible to put an order of magnitude estimate on the cavity-wide mean vertical heat flux, given estimates of inflow/outflow temperatures, residence time, melt rate, etc. That would put the numbers quoted in the abstract into a useful context.

Thank you for this suggestion. We have modified our abstract to read:

"Average vertical heat fluxes are on the order of 0.1 $W\ m^{-2}$ and maximum heat fluxes reach 52 $W\ m^{-2}$. This is compared to the 59 $W\ m^{-2}$ to 176 $W\ m^{-2}$ needed to maintain observed average basal melt rates at DIS. Turbulent mixing is higher in the fast-flowing inflow region and over rough topography. We show a highly complex spatial pattern of turbulent mixing and of bottom topography. The bottom topography is currently not resolved in bathymetry products and both the topography and turbulent mixing are currently not resolved in models of ice-shelf–ocean interactions. The levels of turbulent mixing experienced by the warm mCDW inflow to the DIS will lead to negligible loss of heat during its path to the grounding line, leaving plenty of heat available to melt the ice shelf base there. Higher average vertical heat fluxes than observed here must occur in areas of the cavity not resolved in this study. "

We have expanded on your suggestion and added the following to our results
section:

"We can estimate the DIS basal melt, assuming that the entire heat flux is
used to melt ice at a depth of approximately 1000 m. With this assumption
the melt rate $m$ is

$$m = \frac{Q_t}{L_i \rho_i} \tag{1}$$

with $L_i = 3.315 \times 10^5 \, \text{J kg}^{-1}$ the latent heat of fusion at 1000 dbar, and
$\rho_i = 917 \text{kg m}^{-3}$ the density of ice. This results in melt rate estimates of
$2 \times 10^{-3} \, \text{m yr}^{-1}$ to $11 \times 10^{-3} \, \text{m yr}^{-1}$. Published estimates for area averaged
melt rates under DIS range from $6\,1 \, \text{m yr}^{-1}$ to $18\,3 \, \text{m yr}^{-1}$ (Gourmelen et al.,
2017; Lilien et al., 2018; Robertson, 2013; Jenkins et al., 2018; Schodlok et
al., 2012) with some estimates up to $32\,9 \, \text{m yr}^{-1}$ (Jenkins et al., 2018). The
low upward heat flux within the mCDW layer is thus not able to maintain
the observed melt rates under DIS. To achieve the melt rate estimates from
(Gourmelen et al., 2017; Lilien et al., 2018; Robertson, 2013; Schodlok et al.,
2012; Jenkins et al., 2018) the vertical heat flux would need to he greater than
$59 \, \text{W m}^{-2}$ to $316 \, \text{W m}^{-2}$, values three to four orders of magnitude larger than
our median estimates and up to six times our maximum estimate (Table 2).

Davis et al. (2025) showed elevated levels of $\varepsilon$ in the ice–ocean boundary
layer under Thwaites, and Kimura et al. (2016) observed elevated values of $\varepsilon$
close to the ice–ocean interface and over a bathymetric ridge in front of the
PIIS grounding line. In these areas high turbulent kinetic energy dissipation
rate and high vertical and horizontal temperature gradients lead to high
temperature fluxes. Our study did not reach the ice–ocean boundary layer
or the ridge limiting flow to the DIS grounding line (Figure 1) which may
explain the underestimate of the area averaged ice shelf melt rate using the
observed heat fluxes. The value for $\kappa$ for stably stratifies water, used in the

ISOMIP+ protocol, matches our estimate of $\kappa$. Thus, modelled vertical heat
transport, in regions for which this estimate is used, could also be too low to
explain observed ice shelf melt rates. The low heat fluxes in the interior of ice
shelf cavities would need to be offset by higher heat fluxes at the grounding
line and in the ice–ocean boundary layer.

We can additionally estimate a melt rate for DIS from the temperature dif-
ference between the inflow and the outflow of the cavity and the average
residence time of water within the cavity. We take the heat needed to warm
the ice shelf to the freezing point temperature and the heat needed to warm
the melt water to the temperature of the outflow into account. A back-of-
the-envelope calculation for melt rate gives:

$$m = \frac{V_{in}C_p\bar{\rho}\left(\theta_{in} - \theta_{out}\right)}{\rho_i A_{DIS}\left(C_i\left(\theta_f - \theta_{ice}\right) + L_i + C_p\left(\theta_{out} - \theta_f\right)\right)} \tag{2}$$

with $V_{in} = v_{cavity}A_{inflow}$, the volume transport in the inflow; $v_{cavity} = \frac{D}{t}$ the
velocity of the inflow; $D$ the distance water has to travel from the ice front
to the grounding line and back; $t$ the time the water takes to travel to the
grounding line and back; $A_{inflow}$ the area through which water flows into the
cavity; $A_{DIS}$ the area of the DIS; $C_i$ the specific heat capacity of ice at $-2\,°\text{C}$
and $1000\,\text{db}$; $\theta_f$ the freezing point temperature of seawater; $\theta_{in}$ the average
temperature of the inflow to DIS; $\theta_{out}$ the average temperature of the outflow
from DIS; $\theta_{ice}$ the far-field internal temperature of the DIS.

We assume the following values for these parameters: $D = 240\,\text{km}$ (Figure 1);
$t = 2\,\text{months}$ (Milillo et al., 2022; Yang et al., 2022; Girton et al., 2019);
$A_{inflow} = 500\,\text{m} \times 15\,\text{km}$ (solid box in Figure 3); $A_{DIS} = 5200\,\text{km}^2$ (Lilien
et al., 2018); $\theta_f \approx -2\,°\text{C}$; $\theta_{in} = 0\,2\,°\text{C}$ (the average temperature in the solid
box in Figure 3); $\theta_{out} = 0\,17\,°\text{C}$ (the average temperature in the dashed box
in Figure 3, the outflow extends to shallower depths in the water column
than the inflow due to the thinner ice shelf draft in the west of the DIS
(e.g. A. Wåhlin et al., 2024).); $\theta_{ice} = -25\,°\text{C}$, an estimate of the far-field ice
temperature. Our estimate of melt rate and heat flux from in and outflow temperatures is most sensitive to the area over which we average outflow temperatures (Figure 3) and represents an order of magnitude estimate only.

Equation 2 results in an estimate of the melt rate of $\sim 10 \pm 5\,\mathrm{m\,yr^{-1}}$, which lies within the range of published values (e.g. Gourmelen et al., 2017; Lilien et al., 2018; Robertson, 2013; Schodlok et al., 2012; Jenkins et al., 2018). To maintain this melt rate the vertical heat flux in the cavity would need to be $100 \pm 50\,\mathrm{W\,m^{-2}}$, about three orders of magnitude higher than the median values along the east dive track (Table 2). Rearranging Equation 2 allows us to estimate the percentage of the heat entering the ice shelf cavity that is used to melt ice. We estimate that the inflow transports $4 \pm 2\,\mathrm{TW}$ into the cavity and the melt takes up $0\,6 \pm 0\,4\,\mathrm{TW}$, thus, only $\sim 15 \pm 9\,\%$ of the heat entering DIS is used to melt the ice shelf. Modelling studies have estimated this value to be smaller, at 8 % (Jourdain et al., 2017), but within our error range. Transport calculations by Jenkins et al. (2018) yield the same range for heat flux into the cavity as our estimate does, however, their calculated melt rate, derived from melt water fluxes, is significantly higher $(6\,\mathrm{m\,yr^{-1}} - 33\,\mathrm{m\,yr^{-1}})$. These melt rates would require heat fluxes of $60\,\mathrm{W\,m^{-2}} - 317\,\mathrm{W\,m^{-2}}$. We need significantly more measurements under ice shelves to understand the role of mixing in different areas and regimes, and its effect on ice shelf melt rate."

Introduction:

The first paragraph talks about the DIS contribution to Amundsen Sea "mass loss", suggesting that the term refers to shrinkage of the ice sheet. However, the latter part of the paragraph partitions "mass loss" for the ice shelves between calving and melting. In this instance the term does not refer to shrinkage of the ice shelves, but the contribution to the wastage side of the mass budget. Those are different concepts, and the distinction should be clarified.

Thank you for pointing out this ambiguity, we have rephrased this paragraph to now only refer to ice sheet mass loss and basal melt/thinning of the ice shelf. The revised paragraph now reads "Between 1979 and 2017 DIS contributed 0 6 mm to global eustatic sea level rise (Rignot et al., 2019). The rate of discharge across its grounding line has increased throughout the satellite record (Rignot et al., 2019; Mouginot et al., 2014) and the grounding line has retreated (Rignot et al., 2014; Scheuchl et al., 2016; Milillo et al., 2022). The increased ice flux across the Dotson grounding line, coupled with the stable ice flux across the calving front (Rignot et al., 2013; Mouginot et al., 2014) and the increased thinning of the ice shelf (Rignot et al., 2013; Mouginot et al., 2014; Gourmelen et al., 2017; Greene et al., 2022) leads to the conclusion that ocean thermal forcing has increased basal melt of the ice shelf (e.g. Mouginot et al., 2014). Dotson has thinned at a 37% higher rate than the average rate of thinning in the Amundsen Sea (Paolo et al., 2015). "

The last paragraph states that there have been only two previous published studies of mixing beneath ice shelves, but that overlooks studies based on borehole data. The oversight is repeated in other parts of the manuscript.

This is a good point. We clarify the Introduction, by referring to autonomous vehicles: "To our knowledge, there exist two published studies of mixing in an ice-shelf cavity measured by an underwater vehicle, one under Pine Island Glacier (Kimura et al., 2016), and one under the Filchner Ronne Ice Shelf (Davis et al., 2022). We present a third such study, targeting DIS." and discuss comparisons with borehole results in the Discussion section. To our knowledge there are three published studies of successful measurements of mixing through ice shelf boreholes in the Antarctic (Davis & Nicholls, 2019; Venables et al., 2014; Davis et al., 2025), but if you are aware of others we have missed, we would happily include them. We have added the new Figure 8 and the following paragraph to the revised manuscript:

"The highest levels of turbulent mixing occur in the inflow region at the ice front and in the east dive track, decreasing into the cavity (Figure 5 and Figure 6). The east dive track clearly shows the highest values for $\varepsilon$ of the three ALR dive tracks at DIS (Figure 8). The range, maximum and median values of $\varepsilon$ measured with the VMP at the ice front are higher than those observed in the cavity with the ALR, but ranges have a wide overlap.

We compare our observations of turbulent kinetic energy dissipation rate with other observations under Ronne Ice Shelf (measured using a MicroRider mounted on an ALR; Davis et al., 2022), George VI Ice Shelf (measured with a VMP through a borehole; Venables et al., 2014), Thwaites Ice Shelf (measured with a VMP through a borehole; Davis et al., 2025) and Larsen C ice shelf (measured with a turbulence instrument cluster moored close to the ice–ocean interface; Davis & Nicholls, 2019). The distributions of $\varepsilon$ under Ronne and George VI have similar shapes and ranges to our observations (Figure 8). The VMP observations do, however, show much higher maximum values. This is likely caused by the greater vertical extent of the VMP measurements, which reach into the ice–ocean boundary layer where $\varepsilon$ is elevated (Davis et al., 2025). This is confirmed by the measurements 2.5 m and 13.5 m from ice-ocean interface under Larsen C, which show the highest average values of $\varepsilon$ of the measurements included in Figure 8. Further studies are needed to establish whether observed differences between ice shelves are driven by different mixing regimes or different observation techniques. The current state of knowledge leads us to conclude that the measurements taken under Dotson agree remarkably well with available distributions of $\varepsilon$ from other ice shelves, outside of the ice–ocean boundary layer. "

Data and methods:

On line 127 there is a parenthetical note to authors that has not been addressed.

We are very sorry to have missed that! Thank you for pointing it out, the note has been removed.

On line 140 the dimensionless parameter could more precisely be referred to as a "gradient Richardson number".

Yes, we will make that change.

On line 165 there is a mention of detiding LADCP data using CATS2008. Elsewhere it is stated that tides are unimportant, and CATS cannot be trusted because of the poor bathymetry in the model. One comment refers to (mainly) sub-ice data and the other to ice front data, but nevertheless the treatment seems inconsistent. If bathymetry is poor beneath the ice, won't that influence currents at the ice front? If tides are weak enough to be ignored, why bother with detiding the LADCP data?

Thank you, we acknowledge that this is confusing for readers and have rephrased the text to make this clearer.

- As noted by the reviewer, the bathymetry is worse in the cavity than outside it due to a lack of observations in the cavity (as shown in Figure 7), which is why we trust the CATS2008 model more at the ice front than in the cavity.

- At the ice front section we have ship ADCP for validation of the CATS model, making us more confident in the tide model solution for correcting the LADCP.

- The tidal currents at the ice front are indeed small O(1 $cms^{-1}$) (Dotto et al., 2025).

- We expect the effect of tides in the cavity to be influenced by the barotropic jump at the ice front, something not well captured in the CATS model.

- We could not identify a tidal signal in our ALR ADCP time series in the cavity and thus we concluded that the error introduced by a faulty tidal model would likely be larger that the error caused by not detiding.

- On a practical point, the LADCP data were processed and detided by Dotto et al. (2025), and in order to stay consistent with Dotto et al. (2025) we use their detided dataset (Dotto, Tiago S et al., 2024) in our study.

We make these methodological approaches clearer in the revised paper. The relevant section of the methods now reads:

"Upward-looking and downward-looking LADCP measurements were processed with the LDEO_IX toolbox, incorporating information from the vessel-mounted ADCP, CTD, GPS and bottom track from the LADCP (Thurnherr, 2021). The processed data were averaged into 8-m vertical bins and detided using an updated version of the CATS2008 Antarctic tide model (Padman et al., 2002; Erofeeva et al., 2024). Modelled tidal current components are on the order of $1\,\mathrm{cm\,s^{-1}}$ at the ice front and the tide model agrees well with tides extracted from the shipboard ADCP data (Dotto et al., 2025). Conversely, the ALR ADCP data are not detided due to the ill-constrained bathymetry under DIS, the absence of a detectable tidal signal in a spectral analysis of the ALR ADCP currents in the cavity, and the risk of degrading the ADCP data quality with an ill-fitting tidal model. "

Results and discussion:

In the title of section 3.1 and elsewhere in the manuscript the edge of the ice shelf is referred to as the "ice shelf front". The correct term for that feature is the "ice front".

We have made this change.

In figure 3, is there a "black line" showing the track of the ALR (third line of the caption)? I couldn't see one.

Thank you for this feedback, we have removed the line (which is obscured in large parts by the coloured dots showing the ALR measurements along the track.) and the reference to it from the figure caption.

On lines 222-223 it is stated that water at the ice front is colder and lighter than that in the cavity. Does that refer only to measurements made with ALR? Was the warmer, saltier water apparent in the section observed with the ship? If not, I think it deserves some comment about where that warm, salty water may have come from? Waters in the cavity must be cooled and freshened, so the observation must say something about variability at the ice front. If, on the other hand, an equally warm, salty water mass is present in the ship CTD data, then the statement in the paper is a little misleading.

The statement on lines 222–223 refers to the ship CTD section along the ice front. We have made this clearer in the revised paper. The revised paragraph reads:

"Water at the ice front (measured with the ALR and the ship CTD) is colder but lighter than water found deeper in the cavity (Figure 6). The temperature (Figures 6 and 5) and salinity (not shown) in the cavity generally increase with depth. The presence of warmer, saltier, and denser water in the cavity than at the ice front may indicate seasonal or interannual variability in the properties of the water at the ice front (as described by Kim et al. (2021)) and thus of water flowing into the cavity."

On lines 228-229, and elsewhere, it is stated that the observations reported in the paper are important for establishing mixing rates that can be "incorporated into numerical models". It is not clear to me how these data would be incorporated into a model. Perhaps the point could be clarified?

Thank you, we are happy to expand and clarify how our results may inform modelling efforts. We have removed the reference to parameterisations in lines 228–229 and adding the following to the manuscript at the end of section 3.2: "Maximum and median values of diapycnal diffusivity $\kappa$, vertical heat flux $Q_T$, and vertical salt flux $Q_S$ from our observations under DIS are given in Table 2. Our median values of diapycnal diffusivity ($O(10^{-4}\,\mathrm{m^2\,s^{-1}}$–$O(10^{-5}\,\mathrm{m^2\,s^{-1}})$) are the same order of magnitude as globally-averaged ocean values (Waterhouse et al., 2014). The maximum values of diapycnal diffusivity in our study ($O(10^{-2}\,\mathrm{m^2\,s^{-1}}$–$O(10^{-3}\,\mathrm{m^2\,s^{-1}})$) match values observed close to the seabed over rough terrain or at ridges (Waterhouse et al., 2014).

Our observations under DIS provide valuable metrics against which turbulent mixing processes in numerical models could be assessed. Turbulent kinetic energy dissipation dissipation is not modelled or parameterised in regional or global models. Instead, diapycnal diffusivity $\kappa$ is parametrised. A common parametrisation of diapycnal diffusivity in ice shelf cavities is the vertical profile method from Large et al. (1994) (e.g. in ROMS; Gwyther et al. (2015) or MITgcm; Nakayama et al. (2017)) which assumes higher values of $\kappa$ in boundary layers than in the interior. The interior mixing is made up of contributions from internal waves (parameterised as a constant), from shear instability (parameterised from the gradient Richardson number), and from double diffusion (parameterised from the double diffusion density ratio) (Large et al., 1994). The ice base roughness has been shown to influence the ice–ocean boundary layer mixing and the heat and salt flux into the boundary layer, and thus the spatial and temporal distribution of ice shelf melt (Gwyther et al., 2015). We are not aware of studies investigating the effects of spatially variable bottom boundary layer roughness on mixing and basal melt in an ice shelf cavity. The range of values for $\kappa$, the spatial variability, and forcing mechanisms we discuss, can be compared to the values and variability of the $\kappa$ profile parametrisation. This may allow a better understanding of the contribution of different drivers to mixing and of how realistic model mixing is.

Another common choice to parametrize mixing, used in the ISOMIP+ protocol (Asay-Davis et al., 2016), is to prescribe constant values for $\kappa$ in the vertical and horizontal, with higher values where the water column stratification is unstable. In stably stratified water, as under DIS, the ISOMIP+ protocol sets as $\kappa_{v,stable} = 5 \times 10^{-5} m^2 s^{-1}$ (Asay-Davis et al., 2016). The value of $\kappa$ used in ISOMIP+ has the same order of magnitude as the median value in the centre_short dive track ($2 \times 10^{-5} m^2 s^{-1}$), but is an order of magnitude lower than the median $\kappa$ on the east dive track ($1.1 \times 10^{-4} m^2 s^{-1}$) and 2–3 orders of magnitude lower than the maximum values we find within the cavity (Table 2). Thus, the constant value of $\kappa$ used in ISOMIP+ is a good choice for slow flows with low shear over smooth topography, but may underestimate mixing in other areas which may in turn influence modelled ice-shelf melt. "

On line 230, mention is made of a 100 m thick "melt layer" observed through a borehole. What feature are you referring to? The upper 100 m of the borehole data shown in Figure 2 appear to indicate the presence of less meltwater than deeper in the water column. Why is that? A shallow intrusion of WW along the ice shelf base?

You are correct that this water may show a shallow intrusion of WW, we have removed the reference to a melt layer.

Lines 197-203 draw comparisons with observations made at Pine Island Ice Front but point out differences in the physical setting. One difference that might be relevant, but which appears to have been overlooked, is that in the case of Pine Island there is a neighboring ice shelf to the north, so the northern sidewall of the channel confining the Pine Island Ice Shelf does not extend all the way to the ice front.

Thank you for pointing this out, we have changed the relevant paragraph in the revised manuscript to reads: "At the nearby Pine Island Ice Shelf (PIIS) Naveira Garabato et al. (2017) conducted ADCP and VMP transects along the calving front. Naveira Garabato et al. (2017) do not detect a fast, narrow, turbulent inflow current, unlike what we observed at DIS (Figure 3). High rates of turbulent kinetic energy dissipation below the WW were mostly confined to the PIIS outflow. The PIIS is connected to another ice shelf cavity to the north and may receive some of its inflow from under this neighbouring ice shelf, which may decrease the inflow across the PIIS front and possibly the turbulent mixing there. Additionally, the ice shelf draft of the PIIS is deeper ($\approx 400\,\mathrm{m}$) than the DIS ($\approx 350\,\mathrm{m}$). The ice shelf draft induces a barotropic jump (an abrupt change in water column thickness, blocking flow along constant lines of water column thickness) and limits barotropic inflow to the cavity (A. K. Wåhlin et al., 2020), thus decreasing inflow current velocities and possibly turbulent mixing. "

The last four paragraphs compare findings with other AUV based observations of microstructure beneath ice shelves. However, elsewhere the manuscript highlights the differences between those regions. That makes the discussion feel like one that is motivated by common methodology rather than common physical setting. Why overlook borehole measurements of turbulence that have been made within cavities? Later in the section it is suggested that the

AUV track beneath FRIS is 9 km long, but that does not seem to fit with the figures in the cited paper. At the end the of the section the text again talks about improving parameterisations of mixing in models, but again, I don't really see how you would use the data for that.

Yes, these are good points, thank you for the suggestions. We have strengthened the discussion by adding comparisons with borehole data to the revised manuscript, we have given details on the revised text above. Apologies for the incorrect length of the FRIS dive track, this has been corrected. We have expanded our argument on how our observations can be used to inform modelling studies, the relevant text is included above.

Lines 334-335 suggest that the small vertical heat flux observed means that a lot of ocean heat can be used to melt ice at the grounding line. But how much is used there? The outflows at the ice front remain above the freezing point, so some ocean heat that enters the cavity exits it without being used for melting. Again, can you estimate some global budgets for the amount of heat used for melting and the overall average vertical heat flux that could put your spatially-limited observations in a DIS-cavity-relevant context?

These are good suggestions. We have added an estimation of the heat flux in DIS to the revised manuscript. The paragraph reads: "
[revised manuscript text omitted]

Rignot, E., Jacobs, S., Mouginot, J., & Scheuchl, B. (2013, July). Ice-Shelf Melting Around Antarctica. *Science*, *341*(6143), 266–270. doi: 10.1126/ science.1235798

Rignot, E., Mouginot, J., Morlighem, M., Seroussi, H., & Scheuchl, B. (2014, May). Widespread, rapid grounding line retreat of Pine Island, Thwaites, Smith, and Kohler glaciers, West Antarctica, from 1992 to 2011. *Geophysical Research Letters*, *41*(10), 3502–3509. doi: 10.1002/2014gl060140

Rignot, E., Mouginot, J., Scheuchl, B., van den Broeke, M., van Wessem, M. J., & Morlighem, M. (2019, January). Four decades of Antarctic Ice Sheet mass balance from 1979–2017. *Proceedings of the National Academy of Sciences*, *116*(4), 1095–1103. doi: 10.1073/pnas.1812883116

Robertson, R. (2013, June). Tidally induced increases in melting of Amundsen Sea ice shelves. *Journal of Geophysical Research: Oceans*, *118*(6), 3138–3145. doi: 10.1002/jgrc.20236

Scheuchl, B., Mouginot, J., Rignot, E., Morlighem, M., & Khazendar, A. (2016, August). Grounding line retreat of Pope, Smith, and Kohler Glaciers, West Antarctica, measured with Sentinel-1a radar interferometry data. *Geophysical Research Letters*, *43*(16), 8572–8579. doi: 10.1002/2016gl069287

Schodlok, M. P., Menemenlis, D., Rignot, E., & Studinger, M. (2012). Sensitivity of the ice-shelf/ocean system to the sub-ice-shelf cavity shape measured by NASA IceBridge in Pine Island Glacier, West Antarctica. *Annals of Glaciology*, *53*(60), 156–162. doi: 10.3189/2012aog60a073

Thurnherr, A. M. (2021). *How To Process LADCP Data With the LDEO Software (Version IX.14)*. Retrieved from *https://www.ldeo.columbia.edu/~ant/LADCP.html*. https://www.ldeo.columbia.edu/ ant/LADCP.html.

Venables, E., Nicholls, K., Wolk, F., Makinson, K., & Anker, P. (2014). Measuring turbulent dissipation rates beneath an Antarctic ice shelf. *Marine Technology Society Journal*, *48*(5), 18–24. doi: 10.4031/mtsj.48.5.8

Wåhlin, A., Alley, K. E., Begeman, C., Hegrenæs, Ø., Yuan, X., Graham, A. G. C., . . . Heywood, K. J. (2024, August). Swirls and scoops: Ice base melt revealed by multibeam imagery of an Antarctic ice shelf. *Science Advances*, *10*(31). doi: 10.1126/sciadv.adn9188

Wåhlin, A. K., Steiger, N., Darelius, E., Assmann, K. M., Glessmer, M. S., Ha, H. K., . . . Viboud, S. (2020, February). Ice front blocking of ocean heat transport to an Antarctic ice shelf. *Nature*, *578*(7796), 568–571. doi: 10.1038/s41586-020-2014-5

Waterhouse, A. F., MacKinnon, J. A., Nash, J. D., Alford, M. H., Kunze, E., Simmons, H. L., . . . Lee, C. M. (2014, July). Global Patterns of Diapycnal Mixing from Measurements of the Turbulent Dissipation Rate. *Journal of Physical Oceanography*, *44*(7), 1854–1872. doi: 10.1175/jpo-d-13-0104.1

Yang, H. W., Kim, T.-W., Dutrieux, P., Wåhlin, A. K., Jenkins, A., Ha,
H. K., . . . Cho, Y.-K. (2022, March). Seasonal variability of ocean cir-
culation near the Dotson Ice Shelf, Antarctica. *Nature Communications*,
*13*(1). doi: 10.1038/s41467-022-28751-5

---

## Author Comment (AC4)

**Response to reviewer R3 comments**

August 22, 2025

Thank you for your detailed and helpful review. In this document, reviewer comments are in **black** and our comments are in **red**. New text added to the manuscript is in blue.

This paper presents an interesting set of observations, in an environment difficult to access. The analysis is solid. It would be good to put these observations in the context of the previous work that has been done around Dotson - I understand that there are little observations in the cavity, but are the conditions along the face 'unusual'? It's hard to tell, and I acknowledge that this is not about long-term observations at Dotson, but it would be useful to put these observations in a broader context.

Thank you for your positive review of our manuscript. We have added a sentence that the ice front properties we observed in 2022 are within the usual range: "The temperature and salinity at the ice front are within the historic range of watermass distributions and properties at DIS (Kim et al., 2021)."

As pointed out by the other reviewers, some of the key results are a bit either overstated, or unclear.

We have addressed the concerns of the other two reviewers in our responses, which includes modifying some of our key points. The precise changes are detailed in the responses to reviewer 1 and 2.

For example, on L193, one would be hard-pressed to directly identify the 'enhanced mixing at the inflow' and it being over a larger area than that of the outflow from the section alone - it might useful to show a profile or two of dissipation rates. Sampling (station spacing) might be important when talking about "area", which is not discussed here.

We have changed the paragraph you refer to. It now reads:"Below $500\,\mathrm{m}$ depth, turbulent kinetic energy dissipation is elevated in the inflow (compared with other areas below $500\,\mathrm{m}$ along the ice front). Turbulent kinetic energy dissipation is $\approx 10^{-8}\,\mathrm{W\,kg^{-1}}$ in the inflow over an area approximately $7\,\mathrm{km}$ wide and $200\,\mathrm{m}$ high (Figure 3d; turbulent kinetic energy dissipation rate is elevated between 38 km and 45 km of the ice front and $\sim 200\,\mathrm{m}$ above the seabed). "

Overall, I don't have many comments that were not captured by the other reviewers. This is an interesting paper and it should be published.

Thank you for your positive review.

**References**

Kim, T., Yang, H. W., Dutrieux, P., Wåhlin, A. K., Jenkins, A., Kim, Y. G., . . . Cho, Y. (2021, December). Interannual Variation of Modified Circumpolar Deep Water in the Dotson-Getz Trough, West Antarctica. *Journal of Geophysical Research: Oceans*, *126*(12). doi: 10.1029/2021jc017491

---

## Author Response (AR2)

**Response to reviewer comments**

September 21, 2025

Thank you for your detailed and helpful review. In this document, reviewer comments are in **black** and our comments are in **red**. New text added to the manuscript is in blue.

**1 Reviewer 1**

Thank you for your responses to my suggestions. I am happy with the responses with one exception: I apologize my comment on the finite resolution of the ADCP data was not clear. I was not referring to its use in computing fine scale parameterization but rather in its use in computing Richardson number. Because the instrument response causes small scale motions to be smoothed (Polzin 2002), shear is underestimated significantly and hence Richardson number is biased high. Hence, the value estimated from the ADCP is not comparable to the stability value of 1/4. Additionally, as noted elsewhere, Ri is computed with in-situ shear but only a single N2 profile, and so is not complete. For these reasons (plus that my eye does not see a strong correlation between epsilon and low Ri regions in the authors' data) I suggested that this section be toned down by i) using Ri as a qualitative indicator and removing reference to where the estimated (high-biased) values are ¡1/4, and i) toning down the claims that there is strong agreement between epsilon and low Ri.

Thank you for your clarification. We have removed references to Ri < 1/4

from the results and discussion. The methods section now adds the following to the introduction of Ri:

Our values of Ri are biased high because the ADCP underestimates vertical shear (Polzin et al., 2002), thus we will confine our discussion of Ri to relative values.

The results and discussion of RI now reads: The region of high turbulent kinetic energy dissipation rate $\varepsilon$ in the inflow (Figure 3d) coincides with instances of low Ri captured at $40\,\mathrm{km}$ (Figure 3h). Turbulent kinetic energy dissipation rate is larger than $10^{-8}$ here, one to two orders of magnitude higher than the background value (Figure 3d). Dotto et al. (2025) found similar results for the outflow of DIS. Although areas of high $\varepsilon$ extend beyond areas of low Ri, $\varepsilon$ is higher and Ri is lower in the upper watercolumn and close to the seabed. We observe areas of low Ri that are not associated with high values of $\varepsilon$, e.g. at $25\,\mathrm{km}$ along the transect.

We have removed references to Ri from our discussion of correlations. The relevant sentences now read: They also coincide with areas of high vertical current shear and high along slope velocity (Figures 9 and 7).......Ri is low in the area of high turbulent kinetic energy dissipation rate observed along the east dive track at $-1\,\mathrm{km}$ from the ice front (Figure 9).

**2 Reviewer 2**

In this revision, the authors have thoroughly and thoughtfully addressed my concerns with their earlier draft. As a result, I think the manuscript is improved to the point that it is acceptable more-or-less as is. I did, however, spot a number of issues, all rather minor and mostly typographical in nature, and I think those should be addressed before the paper is formally accepted. Those remaining issues are listed below.

Lines 7-10: I note that, in response to Reviewer 1, the authors changed to a non-italic font for units. However, the units are italicised in the abstract. I

agree with Reviewer 1; non-italic throughout is best. Thank you for pointing this out, we have corrected this in the abstract.

Line 16: "... areas of the cavity not accessed during this study." Thank you, this change has been made.

Line 40 (and elsewhere): The flow along the ice shelf base is referred to as a "buoyant plume". While plume theory is often used as the basis for reduced-physics models, it is not an accurate description of reality. A better description might be a "buoyant current". Thank you, we have changed all instances of plume to current.

Line 41: "... modifying the properties of both the inflowing water, which ultimately interacts with ice near the grounding line, and water carried by the buoyant current out of the cavity." Thank you, this change has been made.

Line 60: "... dense mCDW inflow, ...". Thank you, this change has been made.

Line 87: "... at 1 s-1 [or 1 Hz if you prefer] frequency ... " or "... at 1 s intervals ...". Thank you, this change has been made.

Line 99: "... microstructure data from the ALR were processed ...". Thank you, this change has been made.

Line 126: "... is a measure of the vertical mixing of ...". Thank you, this change has been made.

Line 240: I agree with Reviewer 1. "Barotropic jump" is not an "oceano-graphic term for an abrupt change in water column thickness". A search for the word "jump" in the cited Wåhlin paper brings up no occurrences. Why not just stick with "The ice shelf draft induces an abrupt change in water column thickness, blocking flow along isolines of water column thickness, and thus limits barotropic inflow to ...".

Thank you, we have adopted your preferred wording.

Lines 257-259: Or possibly throughflow from beneath Crosson Ice Shelf?

Possibly, but this has not been well studied. We have added a reference to a throughflow nonetheless. The additional sentence reads: Alternatively, warmer water might be able to enter the DIS cavity from the neighbouring Crosson Ice Shelf cavity (indications of a deep connection are described in (Girton et al., 2019), however, they observed flow from DIS to Crosson)

Line 262: "... a valuable addition to our knowledge ..."

Thank you, this change has been made.

Line 281: "... from the ice-ocean interface ...".

Thank you, this change has been made.

Line 348: "... to navigate a step in water column thickness ...".

Thank you, this change has been made.

Line 349: "... increased rates of turbulent kinetic ...".

Thank you, this change has been made.

Line 352: "...than the ice front draught."

Thank you, we have changed the text, but we think it should be 'ice front draft'?.

Line 356: "... Bay, found ...".

Thank you, this change has been made.

Line 377: "... kinetic energy dissipation is not commonly modelled ...". There are examples of - models being used, although they are uncommon.

Thank you, this change has been made.

Line 428: "... lead to high heat fluxes."

Thank you, this change has been made.

Line 430: "... for stably stratified water ...".

Thank you, this change has been made.

Lines 490-491: "...from DIS inflow and outflow temperatures agree with published ranges of ice shelf melt rates ...".

Thank you, this change has been made.

Line 496: There is a parenthetical question mark, which presumably should be deleted.

Thank you, we have corrected the citation error that led to this.

Line 512: "... for the use of ...".

Thank you, this change has been made.

**References**

Dotto, T. S., Sheehan, P. M. F., Zheng, Y., Hall, R. A., Damerell, G. M., & Heywood, K. J. (2025, May). Heterogeneous Mixing Processes Observed in the Dotson Ice Shelf Outflow, Antarctica. *Journal of Geophysical Research: Oceans*, *130*(5). doi: 10.1029/2024jc022051

Girton, J. B., Christianson, K., Dunlap, J., Dutrieux, P., Gobat, J., Lee, C., & Rainville, L. (2019). Buoyancy-adjusting Profiling Floats for Exploration of Heat Transport, Melt Rates, and Mixing in the Ocean Cavities Under Floating Ice Shelves. In *Oceans 2019 mts/ieee seattle* (p. 1-6). doi: 10.23919/OCEANS40490.2019.8962744

Polzin, K., Kunze, E., Hummon, J., & Firing, E. (2002). The finescale response of lowered ADCP velocity profiles. *Journal of Atmospheric and Oceanic Technology*, *19*(2), 205–224. doi: 10.1175/1520-0426(2002)019⟨0205:tfrola⟩2.0.co;2